# Development of schemas revealed by prior experience and NMDA receptor knock-out

**George Dragoi[1,2,3,4]\*, Susumu Tonegawa[1,2,3,4]\***

[1]The Picower Institute for Learning and Memory, Massachusetts Institute of Technology, Cambridge, United States; [2]RIKEN-MIT Center for Neural Circuit Genetics, Massachusetts Institute of Technology, Cambridge, United States; [3]Department of Biology, Massachusetts Institute of Technology, Cambridge, United States; [4]Department of Brain and Cognitive Sciences, Massachusetts Institute of Technology, Cambridge, United States

**Abstract** Prior experience accelerates acquisition of novel, related information through processes like assimilation into mental schemas, but the underlying neuronal mechanisms are poorly understood. We investigated the roles that prior experience and hippocampal CA3 $N$-Methyl-D-aspartate receptor (NMDAR)-dependent synaptic plasticity play in CA1 place cell sequence encoding and learning during novel spatial experiences. We found that specific representations of de novo experiences on linear environments were formed on a framework of pre configured network activity expressed in the preceding sleep and were rapidly, flexibly adjusted via NMDAR-dependent activity. This prior experience accelerated encoding of subsequent experiences on contiguous or isolated novel tracks, significantly decreasing their NMDAR-dependence. Similarly, de novo learning of an alternation task was facilitated by CA3 NMDARs; this experience accelerated subsequent learning of related tasks, independent of CA3 NMDARs, consistent with a schema-based learning. These results reveal the existence of distinct neuronal encoding schemes which could explain why hippocampal dysfunction results in anterograde amnesia while sparing recollection of old, schema-based memories.

**\*For correspondence:** gdragoi@mit.edu (GD); tonegawa@mit.edu (ST)

**Competing interests:** The authors declare that no competing interests exist.

**Reviewing editor**: Howard Eichenbaum, Boston University, United States

## Introduction

An essential capacity of the brain is to form internal representations of the external world. Whereas representations of past experiences can be stored internally and be rapidly recalled from memory, neuronal representations of novel experiences develop from the interaction between specific external stimuli and the spontaneous internal neuronal dynamics during the novel experience. During the free exploration of spatial environments, individual hippocampal neurons fire at specific spatial locations of the animal and are activated sequentially along the trajectory of the animal (place cells) (*O'Keefe and Dostrovsky, 1971*; *O'Keefe and Nadel, 1978*; *McNaughton et al., 1983*; *O'Keefe and Recce, 1993*; *Wilson and McNaughton, 1993*; *Lee and Wilson, 2002*). Sequences of place cells with partially overlapping fields fire with compressed temporal delays that correspond to the Euclidian distance between the location of their place field peaks (*Muller et al., 1996*; *Dragoi and Buzsaki, 2006*). This phenomenon, known as sequence compression (*Skaggs et al., 1996*; *Dragoi and Buzsaki, 2006*) is thought to be an animal model of the internal representation of an external space by the place cell assemblies in the CA1 area of the rodent hippocampus (*Skaggs et al., 1996*; *Dragoi et al., 2003*; *Dragoi and Buzsaki, 2006*; *O'Keefe and Nadel, 1978*).

The upstream auto-associative area, CA3, has been proposed to orchestrate (*Tsodyks, 1999*; *Dragoi and Buzsaki, 2006*), together with entorhinal cortex and via synaptic plasticity (*Dragoi et al., 2003*),

**eLife digest** Learning is an inherent feature of the living animals. During development and childhood we learn a large repertoire of items that we encounter for the first time in our life, such as the names of things and how to perform certain tasks. This de novo learning process takes a relatively long time and generally requires repeated exposures to the particular features of the external world that are being learned. Later in life, when we are exposed to novel but related features, we acquire this new information much faster. For instance, if a child learns to associate various odors with various locations around the home—such as associating the smell of bread with the kitchen and the smell air freshener with the bathroom—they will find it easier to make new odor-place associations outside the home, such as associating gasoline with gas stations.

It is thought that the de novo learning process is achieved when the newly acquired information is being consolidated and transferred into long-term memory storage within networks of neurons in the brain. The process of consolidation is believed to lead to the formation of mental schemas that can accelerate learning of novel but related information. Although the concepts of mental schema and related learning are widely used in psychology and education, their underlying neuronal mechanisms are poorly understood.

The formation of new memories depends on a part of the brain called the hippocampus and involves changes in the strength of the connectivity between groups of neurons in a process called synaptic plasticity. In particular, the interaction between a chemical called glutamate, which is released by sender neurons, and proteins called NMDA receptors (which bind the glutamate molecules) on receiver neurons have a central role in synaptic plasticity. Recently, based on experiments with rodents, it has been proposed that the hippocampus is also crucial for the formation of the mental schemas that can accelerate the learning of new spatial association tasks, such as the odor-place associations described above.

Now, Dragoi and Tonegawa reveal that the NMDA receptor in a key subregion of the hippocampus is also involved in the de novo learning of spatial tasks. Using repeated exposures to novel spatial experiences and genetic techniques to block the NMDA receptors in this subregion in mice, Dragoi and Tonegawa discovered that de novo learning involves synaptic plasticity in the hippocampus and, possibly, other regions of the brain. This de novo learning, in turn, enables subsequent spatial learning to be accelerated, even when the NMDA receptors are absent. These results reveal that de novo learning, and related learning processes such as accelerated learning, are underpinned by a number of different mechanisms in the brain, which could help explain why damage to the hippocampus prevents the formation of new memories while preserving other forms of memory and learning.

the functional organization of cellular assemblies (*Hebb, 1949*; *McNaughton et al., 1996*; *Harris et al., 2003*; *Dragoi and Buzsaki, 2006*) in the downstream CA1 region, the source of the hippocampal output to the rest of the cortex.

According to a prevailing model, novel temporal and spatial place cell sequences emerge rapidly in the hippocampus upon exploration of a novel linear track predominantly or exclusively in response to the complex stimuli from the external environment and with minimal or no contribution from the internal neuronal dynamics around the time of the exploration (*Skaggs and McNaughton, 1996*; *Lee and Wilson, 2002*). Subsequently, these sequences are replayed during periods of resting (*Foster and Wilson, 2006*; *Diba and Buzsaki, 2007*; *Davidson et al., 2009*; *Karlsson and Frank, 2009*; *Dragoi and Tonegawa, 2011*) or sleep (*Nadasdy et al., 1999*; *Lee and Wilson, 2002*; *Ji and Wilson, 2007*; *Karlsson and Frank, 2009*) at higher incidences and perhaps facilitate the consolidation of the encoded information (*Girardeau et al., 2009*; *Nakashiba et al., 2009*; *Ego-Stengel and Wilson, 2010*; *Jadhav et al., 2012*).

Recently, we described that temporal sequences of firing that correlated with place cell sequences formed during the exploration of a novel linear track had been expressed during the resting/sleep period preceding the exploration, a phenomenon called preplay (*Dragoi and Tonegawa, 2011*, *2013*). We proposed that spontaneous neuronal activity preceding a novel spatial experience may prime and contribute to the formation of new spatial representations via a neuronal ensemble selection process

(*Dragoi and Tonegawa, 2013*). The dynamics of the interplay between the external stimuli available 'online' during the novel experience and the internal neuronal activity around the time of the experience along with the specific contribution they have to the emergence of a novel spatial representation remain to be elucidated. Here, we investigated the dynamics and contribution of this interplay by studying the development of novel spatial representations in naïve and experienced animals during spatial exploration and their relation to the internal neuronal dynamics preceding and following the new experience.

In order to understand the molecular and cellular mechanisms underlying the development of new spatial representations in the CA1, we studied mice in which experience-dependent N-methyl-D-aspartate receptor (NMDAR)-associated activity and synaptic plasticity were genetically blocked specifically in the upstream auto-associative area CA3 (*Nakazawa et al., 2002*), the source of the main excitatory input and sharp-wave/ripple associated activity into CA1 (*Buzsaki, 1989*; *Nakashiba et al., 2009*). These mutant mice had their CA3 NR1 subunit knocked-out using a Cre/loxP recombination system (*Tsien et al., 1996*; *Nakazawa et al., 2002*) and are referred to as the CA3 NMDAR KO, or simply KO mice. Throughout this study, they were compared with their control littermates, the floxed NR1 mice (*Tsien et al., 1996*; *Nakazawa et al., 2002*), here referred to as control mice (CT). Previous studies using these two groups of mice have shown that deletion of the NR1 subunit in the CA3 area abolished the NMDAR currents in the CA3 pyramidal cells, but not in the downstream CA1 and upstream dentate gyrus area neurons (*Nakazawa et al., 2002*), which resulted in impaired memory acquisition of one-time experiences (*Nakazawa et al., 2003*). The lack of post-synaptic NMDAR function has been shown to block the induction of long-term potentiation of synaptic transmission in the hippocampus (*Bliss and Lomo, 1973*; *Bliss and Collingridge, 1993*; *Tsien et al., 1996*) and other brain areas, a mechanism proposed to underlie learning and memory (*Morris et al., 1986*; *Tsien et al., 1996*; *Moser et al., 1998*).

In order to explore the role of prior experience on the formation of novel spatial representations and spatial learning, we compared the single and ensemble place cell dynamics as well as the animals' performance on a delayed alternation task (*Ainge et al., 2007*) in naïve and experienced mice. We therefore investigated the temporal development of hippocampal place cell assemblies and the dynamics of behavioral performance in the CT (*Tsien et al., 1996*) and KO mice (*Nakazawa et al., 2002*) during de novo spatial exposure and the effect of this (prior) experience on cell assembly and learning dynamics during subsequent exposures to related novel space and behavioral tasks. De novo neuronal representations of spatial experiences were formed on the framework of the spontaneous network activity preceding the experience (*Dragoi and Tonegawa, 2011*) and were subsequently modified and rapidly stabilized via CA3 NMDAR-dependent activity. This prior experience on novel linear tracks accelerated the encoding of subsequent experiences on additional, contiguous or isolated novel tracks and eliminated or significantly reduced their CA3 NMDAR dependence, respectively. We evaluated the behavioral relevance of these neuronal dynamics by comparing them with the performance of naïve and experienced CT and KO animals on a T-maze delayed alternation task. The presence of CA3 NMDARs facilitated the de novo learning of the alternation task; this prior experience accelerated subsequent learning of a related alternation task in a CA3 NMDAR-independent manner, consistent with principles of a schema-based learning (*Bannerman et al., 1995*; *Tse et al., 2007*; *McKenzie et al., 2013*). Moreover, prior experience and CA3 NMDARs modulated the learning rate and the neuronal ensemble dynamics across experimental conditions in a correlated manner, suggesting that these neuronal dynamics, and in particular the expression of stable cell assemblies in CA1, might be part of the neural mechanisms of learning in naïve and experienced animals.

## Results

Ensemble neuronal recordings were performed from the CA1 area of the hippocampus in CT and CA3 NMDAR KO mice alternating between periods of sleep/rest in the sleep/rest box and periods of exploration of portions of walled linear and L-shaped tracks under three conditions of behavioral novelty (*Figure 1A*). Under the de novo condition, naïve mice previously housed in small cages slept in the sleep/rest box (pre-DnRun1 sleep/rest), after which they were allowed to explore a linear track for the first time (i.e., novel track) for two run sessions (DnRun1 and DnRun2) separated by sleep/rest epochs in the sleep/rest box (*Figure 1A*, *Table 1*; 'Materials and methods'). The sleep/rest session between DnRun1 and DnRun2 is referred to as post-DnRun1 when correlated with DnRun1 and pre-DnRun2 when correlated with DnRun2, while the one following DnRun2 is referred to as post-DnRun2. After the

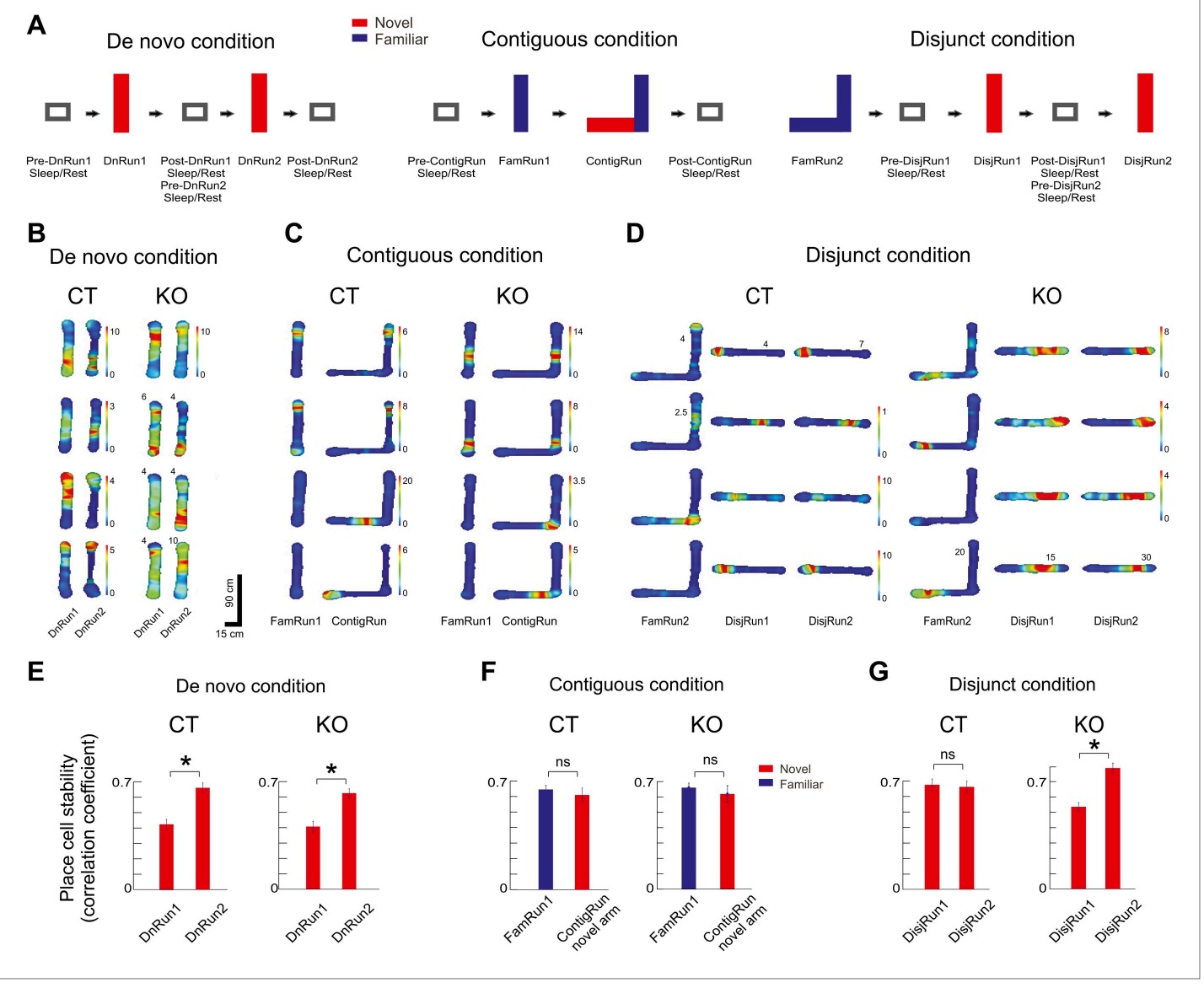

**Figure 1**. Experimental design and place cell dynamics across experimental conditions and sessions. (**A**) Experimental design displaying sleep/rest - run - sleep/rest sessions across de novo, contiguous, and disjunct experimental conditions. (**B–D**) Color-coded firing rate heat maps of place cells across conditions and sessions. For both genotypes, each row displays one place cell. Paired columns represent successive experimental sessions within the same condition and day (de novo, **B**; contiguous, **C**; disjunct, **D**). Colorbars refer to all corresponding sessions, except for those where numbers at the top of the place field maps reflect corresponding peak firing rates. (**E–G**) Increased place map stability with repeated exposure to a novel track. (**E**) De novo condition. (**F**) Contiguous condition. (**G**) Disjunct condition. Stability of the place cell map was calculated as the correlation between place fields at the beginning (first four laps of run) vs the end (last four laps) of the Run sessions. For the contiguous condition ContigRun session, only the place cells active on the novel arm were included. Error bars represent SEM. Stars mark significant differences.

The following figure supplements are available for figure 1:

**Figure supplement 1**. Stereotypy of the mouse behavior (velocity and distance travelled) during the first four and the last four laps of run during different experimental conditions and sessions and across genotypes.

sessions under the de novo condition, the majority of mice were transferred to home cages for one night and returned to the sleep/rest box for the recording session of the next day. This session is referred to as pre-ContigRun session because it preceded a run session under the contiguous condition.

Under the contiguous condition, mice that completed the pre-ContigRun sleep/rest were returned to the linear track they had experienced repeatedly on the previous day. After recordings in the now

**Table 1.** The order and duration of the recording sessions

**Floxed CT mice**

| | De novo condition, duration (min) | | | | | | | Contiguous condition, duration (min) | | | | | | Disjunct condition, duration (min) | | | | | |
|---|---|---|---|---|---|---|---|---|---|---|---|---|---|---|---|---|---|---|---|
| Mouse # | Pre-DNRun1 sleep/rest | DnRun1 | Post-DnRun1 sleep/rest | DnRun2 | Post-DnRun2 sleep/rest | Interval | | fRun | Pre-ContgRun sleep/rest | FamRun1 | ContigRun | Post-ContgRun sleep/rest | Interval | FamRun2 | Pre-DisjRun1 sleep/rest | DisjRun1 | Post-DisjRun1 sleep/rest | DisjRun2 | |
| CT1 | 27 | 60 | 85 | 21 | 34 | ~24 hr | | – | 30 | 26 | 36 | 91 | | | | | | | |
| CT2 | 55 | 54 | 69 | 60 | 30 | – | | – | 69 | 60 | 34 | 56 | ~48 hr | 29 | 96 | 31 | 43 | 15 | |
| CT3 | 52 | 44 | 60 | 31 | 63 | – | | – | 63 | 31 | 34 | 87 | ~48 hr | 22 | 56 | 19 | 48 | 15 | |
| CT4 | 78 | 37 | 31 | 16 | 60 | ~24 hr | | 29 | 62 | 15 | 42 | 81 | ~48 hr | 25 | 50 | 15 | 25 | 16 | |

**CA3 NR1 KO mice**

| | De novo condition, duration (min) | | | | | | | | | Contiguous condition, duration (min) | | | | | | Disjunct condition, duration (min) | | | | |
|---|---|---|---|---|---|---|---|---|---|---|---|---|---|---|---|---|---|---|---|---|
| Mouse # | Pre-DNRun1 sleep/rest | DnRun1 | Post-DnRun1 sleep/rest | DnRun2 | Post-DnRun2 sleep/rest | DnRun3 | Post-DnRun3 sleep/rest | Interval | fRun | Pre-ContgRun sleep/rest | FamRun1 | ContigRun | Post-ContgRun sleep/rest | Interval | FamRun2 | Pre-DisjRun1 sleep/rest | DisjRun1 | Post-DisjRun1 sleep/rest | DisjRun2 |
| KO1 | 24 | 25 | 36 | 21 | 27 | 24 | 117 | ~24 hr | 25 | 14 | 10 | 35 | 83 | ~48 hr | 22 | 53 | 15 | 25 | 16 |
| KO2 | 55 | 38 | 48 | 46 | 60 | 29 | 32 | ~24 hr | – | 30 | 30 | 51 | 62 | | | | | | |
| KO3 | 59 | 48 | 54 | 35 | 30 | – | – | – | – | 30 | 36 | 44 | 48 | ~48 hr | 47 | 69 | 38 | 70 | 21 |
| KO4 | 104 | 50 | 32 | 17 | 28 | – | – | ~24 hr | 36 | 23 | 10 | 40 | 35 | ~48 hr | 36 | 30 | 20 | 35 | 20 |

familiar linear track (FamRun1 session), the wall at one end of the linear track, which had been blocking the access to a second novel linear track attached perpendicular to the first track, was removed. This manipulation allowed mice to run freely on the entire L-shaped track, composed of the new and familiar arms, which constituted the ContigRun session. At the end of this session, the mice were transferred back to the sleep/rest box for post-ContigRun sleep/rest session. A minority of mice went through minor variations of this general protocol (*Table 1*). Two days later, under the disjunct condition, the mice were re-exposed to the now familiar L-shape track (FamRun2 session) after which they were given a session of sleep/rest in the sleep/rest box (pre-DisjRun1). They were then allowed to explore a novel linear track in isolation for two sessions (DisjRun1 and DisjRun2) separated by a sleep/rest session in the sleep/rest box (post-Disj1Run).

## Prior experience and CA3 NMDARs accelerate CA1 place cell tuning and neuronal ensemble temporal correlation in novel environments

For all experimental conditions we determined the across-sessions changes in spatial tuning and stability of individual CA1 place cells (*Hill, 1978*; *Wilson and McNaughton, 1993*; *Nakazawa et al., 2003*; *Frank et al., 2004*; *Cacucci et al., 2007*) and in the lap-by-lap correlation (co-variation) of spiking activity of cell pairs (*Dragoi and Buzsaki, 2006*) during exploration of novel environments. The changes in single cell activity allow the quantification of the temporal development of place fields over successive exploratory sessions (see *Figure 1B–D* for individual cell examples for all three conditions), and the lap-by-lap correlation reflects the organization of neurons in coordinated cellular assemblies whose member pairs exhibit higher temporal correlation than pairs of independent neurons (*Harris et al., 2003*; *Dragoi and Buzsaki, 2006*). The stability of place cell firing within the run session on novel tracks increased with experience in the de novo condition similarly in the CT and KO mice (*Figure 1E*), appeared high from the beginning of the contiguous condition in both genotypes (*Figure 1F*), and appeared high from beginning in the CT mice and increased with repeated experience in KO mice in the disjunct condition (*Figure 1G*).

Under the de novo condition, the spatial tuning of CA1 place cells was relatively poor in session 1 (DnRun1) in both CT (*Cacucci et al., 2007*) and KO mice (*Nakazawa et al., 2003*) by the measure of place field length (*Figure 2A*). Spatial tuning increased faster in CT vs KO mice from session 1 to session 2 (*Figure 2A*, left and 2B left, paired difference in place field length DnRun1–DnRun2, 8 ± 2.3 cm, p<0.003, paired *t* test in CT; 0.6 ± 2.4 cm, p=0.78 in KO; CT vs KO, p<0.027, ranksum test) and within-session DnRun1 (*Figure 2C*, left, spatial tuning, 4.8 ± 2.1 cm, p<0.003 in CT; −0.7 ± 1.5 cm, p=0.3 in KO; CT vs KO, p<0.015, ranksum test). An overnight holding in the home cage led to significantly increased spatial tuning in KO mice the next day (*Figure 2A*, DnRun1 compared to FamRun1, 29.6 ± 2.6 cm compared to 20.6 ± 2.0 cm, p<0.01). The average number of place fields per place cell (>1 Hz) was not different across genotypes and experimental sessions (CT vs KO: 1.52 vs 1.58 fields/place cell in DnRun1, p>0.5, ranksum test; 1.50 vs 1.53 in DnRun2, p>0.7). This result indicates that the experience- and genotype-dependent changes in place field size reported above (*Figure 2B–C*) result from the spatial tuning of initially large place fields rather than from the scattering of large place fields into multiple smaller ones. At the neuronal ensemble level, there was a significant increase in the lap-by-lap correlations with increased experience for both genotypes, although the increase occurred faster in CT compared to KO mice (*Figure 2D*). In the CT, but not KO mice, the increase reached a plateau by the second exploratory session (DnRun2) under the de novo condition (*Figure 2D*, DnRun1 compared to DnRun2, 0.37 ± 0.007 compared to 0.48 ± 0.01, p<10$^{-13}$, ranksum test in CT mice; 0.38 ± 0.005 compared to 0.38 ± 0.004, p=0.38 in KO mice; CT vs KO in DnRun2, p<10$^{-9}$, ranksum test). Although the correlations eventually reached stable CT levels in KO mice, this did not occur until the next day (*Figure 2D*, DnRun1 compared to Fam, 0.37 ± 0.007 compared to 0.48 ± 0.01, p<10$^{-13}$ in CT and 0.38 ± 0.005 compared to 0.48 ± 0.01 in KO, p<10$^{-10}$). These results suggest that the first-time spatial experience on a linear track leads to a gradual increase in spatial tuning and co-variation of ensembles of CA1 place cells encoding the novel experience, and that the temporal dynamics of these processes are facilitated by NMDAR-dependent activity in the auto-associative area CA3 of the hippocampus. The two run sessions of the de novo condition (over 1 hr altogether, *Table 1*) may have provided enough time for the relatively slower non-NMDAR-dependent plasticity to occur and underlie the slower place cell dynamics. The results suggest that CA3 NMDAR-dependent activity and plasticity are specifically involved in the rapid within-session changes in single and ensemble place cell activity.

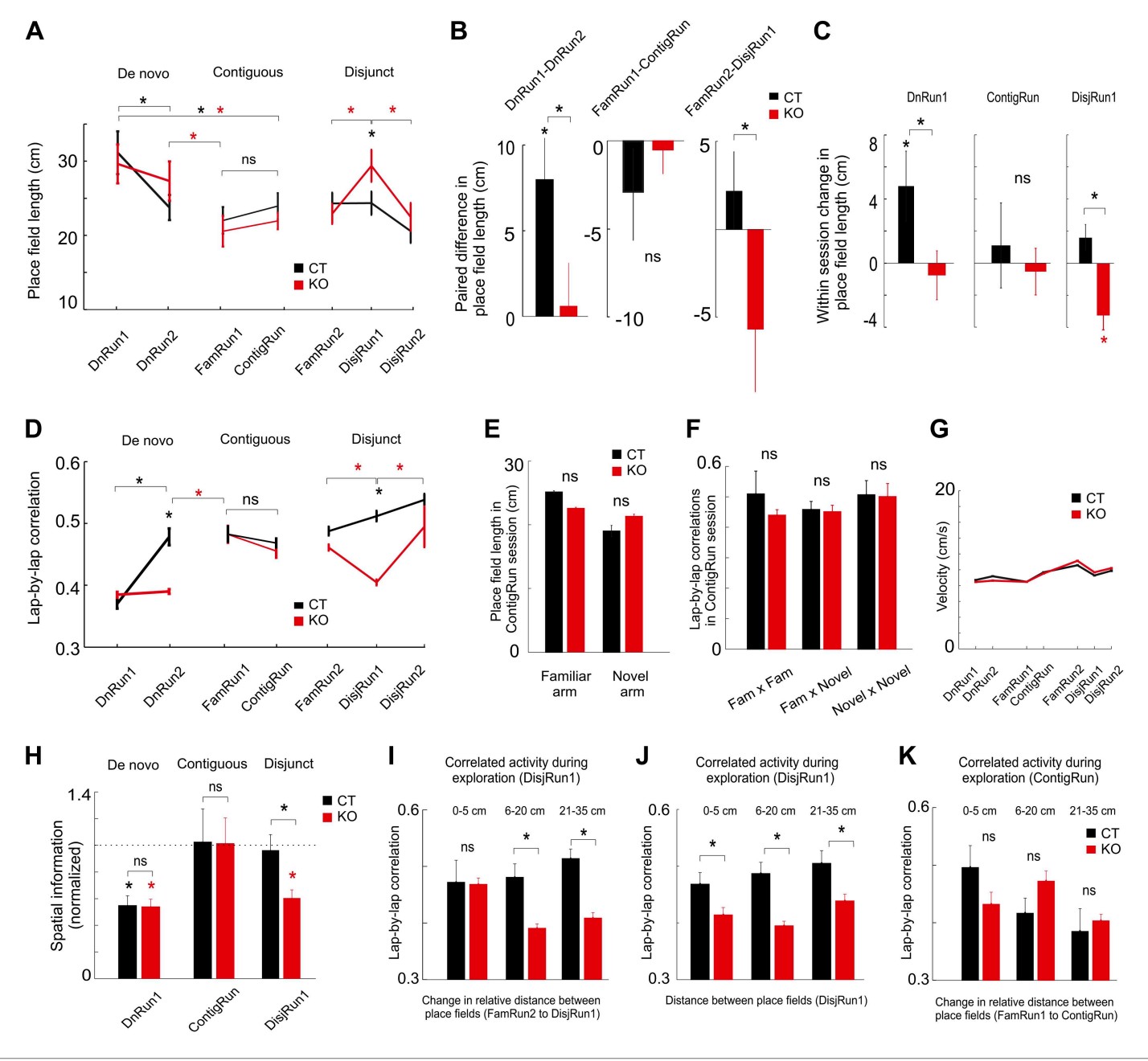

**Figure 2**. Prior experience and CA3 NMDARs accelerate tuning and co-variation of place cells in novel environments. (**A**) Dynamics of place field length for CT (black color) and KO (red color) during de novo, contiguous, and disjunct conditions. (**B**) Paired changes in place field length across the first two sessions for each condition. (**C**) Within-session changes in place field length (first novel session of each condition). (**D**) Dynamics of lap-by-lap correlations of CA1 pyramidal cell spiking activity. The increase in these correlations followed the increase in spatial tuning. (**E**) Comparison between the CA1 place field length in the familiar and novel portion of the L-shape track. (**F**) Comparison of lap-by-lap correlations of spiking activity across place fields in the familiar vs novel portion of the L-shape track. (**G**) Average velocity of mice across different run sessions. (**H**) Average normalized spatial information of place cells during first run sessions on novel track/arm across conditions and genotypes. Dotted line denotes average normalized spatial information during corresponding FamRun sessions used as reference for the spatial information during novel run sessions. (**I–K**) Lap-by-lap co-variation during exploration of the novel track/arm (DisjRun1 in **I** and **J**, ContigRun in **K**) as a function of change in either the relative distance between place fields across the two tracks (disjunct condition in **I**, contiguous condition in **K**) or the distance between place fields in the novel track (**J**). For all subplots, asterisks mark significant differences between groups. Data are from mice CT1-4 and KO1-4.

We next investigated whether the experience acquired during the de novo condition had an effect on a related, but new experience, by evaluating whether the exploration of a novel arm in contiguity with the familiar one (contiguous condition) would re-enact the de novo-type responses only in the novel or in both arms. Surprisingly, for both genotypes, the new place fields were as spatially tuned during the first session (ContigRun) in the novel arm of the L-shaped track as in the contiguous familiar arm (*Figure 2A,E*) and as in the preceding FamRun1 session (*Figure 2A*). No additional spatial tuning could be detected within the ContigRun session within each group of mice (*Figure 2B–C*; FamRun1 compared to ContigRun: 22.1 ± 1.7 cm compared to 23.9 ± 1.7 cm, p=0.46 in CT and 20.6 ± 2.0 cm compared to 21.9 ± 1.1 cm, p=0.34 in KO), across genotypes (CT compared to KO, p=0.6 in FamRun1, p=0.46 in ContigRun), and for both the familiar and the novel portions of the L-shaped track (*Figure 2E*, p>0.05). The average number of place fields per place cell on the L-shaped track was similar across genotypes during ContigRun (CT vs KO, 1.90 vs 1.89, p<0.98). Similarly, the increased lap-by-lap correlations observed in FamRun1 were preserved in the ContigRun session for both groups of mice (*Figure 2D,F*; FamRun1 compared to ContigRun, 0.48 ± 0.01 compared to 0.47 ± 0.009, p=0.41 in CT and 0.48 ± 0.01 compared to 0.46 ± 0.01, p=0.2 in KO), and there were no significant differences in the correlations between genotypes (CT compared to KO, p=0.98 in FamRun1, p=0.42 in ContigRun), nor between the familiar and the novel portions of the L-shaped track (*Figure 2F*, p>0.05). These results suggest that under the contiguous condition, independent of CA3 NMDARs, the spatial information regarding the previously unvisited novel portion of the L-shaped track was rapidly bound with the existing stable representation of the familiar track given the animals' prior experience on the familiar track. These rapid dynamics on the novel arm during the Contig session are in contrast to the slow dynamics of place field tuning and co-variation during de novo exposure to a novel track (i.e., the DnRun1 session for CT and DnRun1–2 sessions for KO). This may reflect the prior recruitment at a subthreshold firing level of the new place cells (*Epsztein et al., 2011*) into stable, tuned cortico-hippocampal cellular assemblies during pre-exposures to the familiar track, which manifested as preplay of the novel arm during the preceding sleep or resting epochs (*Dragoi and Tonegawa, 2011, 2013*).

To determine whether the contiguity between the novel and familiar arms was essential for the rapid spatial tuning and assembly organization and to further assess the effect of experience on novel spatial representations, experienced animals were exposed to an isolated novel linear track in the same general environment (disjunct condition, *Figure 2A*). In this case, exploration of the novel track induced a significant, albeit transient, increase in the CA1 place field length in KO but not CT mice (FamRun2 vs DisjRun1: 23.9 ± 1.4 vs 23.9 ± 1.5 cm in CT, p=0.9; 22.6 ± 1.3 vs 28.9 ± 2 in KO, p<0.01, ranksum test; between genotypes, CT vs KO in DisjRun1, p<0.05; DisjRun1 vs DisjRun2 in KO, p<0.04, *Figure 2A*; paired difference in place field length FamRun2–DisjRun1 CT vs KO, p<0.027, ranksum test, *Figure 2B*; within-session DisjRun1, CT vs KO, p<$10^{-3}$, *Figure 2C*), consistent with an earlier report (*Nakazawa et al., 2003*). The average number of place fields per place cell was slightly higher in CT compared to KO mice, but this relationship was not affected by the additional experience of animals on the novel track (CT vs KO: 1.76 vs 1.52 in DisjRun1 and 1.76 vs 1.47 in DisjRun2, p<0.002). The increased lap-by-lap correlations recorded on the familiar track (FamRun2) decreased with novelty (DisjRun1) in KO but not in CT mice (FamRun2 vs DisjRun1: 0.49 ± 0.006 vs 0.51 ± 0.008, p>0.05 in CT; 0.47 ± 0.005 vs 0.41 ± 0.004, p<$10^{-15}$ in KO; DisjRun1: CT vs KO, p<$10^{-30}$; DisjRun1 vs DisjRun2 in KO, p<$10^{-4}$, *Figure 2D*). The correlations returned to 'familiar levels' by the next session (DisjRun2) in the KO mice more rapidly compared to the de novo condition (compare DisjRun2 to DnRun2 in KO, *Figure 2D*). Overall, these dynamics were not due to differences in the velocity of animal movement across sessions or genotypes (*Figure 2G*), as place field size and lap-by-lap correlations were not significantly correlated with the animal velocity (p>0.11 and p>0.12, respectively). Moreover, the changes in place field tuning occurring within experimental sessions (i.e., DnRun1, DisjRun1, first vs last four laps) were not simply due to a change in the behavior of the mice during the corresponding sessions since their velocity and total distance travelled (path stereotypy) were similar in the first four laps compared with the last four laps of the sessions for both genotypes (first vs last four laps, p>0.05, ranksum test; velocity, *Figure 1—figure supplement 1A*, total distance travelled, *Figure 1—figure supplement 1B*). The changes in lap-by-lap correlations across experimental sessions were not simply a result of changes in animal behavior during the 3s-bin epochs since the distances travelled by the mice on the tracks during the 3s-bins were not correlated with the values of the lap-by-lap correlations calculated over the same timescale (p>0.2). The overall changes in the spatial information of place cells (*Skaggs et al., 1996*) between novel and familiar run sessions for individual CA1 place cells across

genotypes and conditions were consistent with the changes in place field size and in lap-by-lap correlations of pairs of cells (*Figure 2H*).

To further examine the structure of correlations during DisjRun1 exploration, we grouped pairs of place cells active on both FamRun2 and DisjRun1 sessions based on the change in the relative distance between their place field peaks across the two tracks into those with changes within 5 cm (stable assemblies, 8% CT, 10% KO pairs), those with changes between 6 cm to 20 cm (27% CT, 37% KO), and those with changes between 21 cm to 35 cm (32% CT, 26% KO). The remaining pairs, with changes larger than 35 cm were excluded from this analysis. In the CT mice, cells from all three groups displayed relatively high lap-by-lap co-variation during DisjRun1 exploration (*Figure 2I*). In contrast, in KO mice, only cells of the stable assemblies (within 5 cm change) maintained a high lap-by-lap co-variation during DisjRun1, while groups of pairs with changes larger than 5 cm exhibited significantly poorer co-variation (*Figure 2I*, p<0.05, ranksum test). These results suggest a modular, higher order organization of cellular assemblies, in which a certain group of cell pairs (stable assemblies) counters a complete NMDAR-dependent reorganization of the entire neuronal population in response to novel stimuli by maintaining their co-variation independent of CA3 NMDARs. This modular organization of cell assemblies was not simply based on the spatial proximity between place fields on the novel track as co-variations in the KO mice were significantly poorer than in CT mice for all three regimes of changes in place field distance (*Figure 2J*, p<0.05). A similar grouping of the cell pairs active in the contiguous condition during FamRun1 and ContigRun revealed that the changes in cellular ensemble organization upon exploration of the contiguous novel arm are CA3 NMDARs-independent (*Figure 2K*). The large proportion of cell pairs showing >20 cm changes in their spatial relationship from familiar to novel tracks (>65% in CT, >53% in KO mice) indicates that the ensemble of place cells formed distinct representations across the two environments, a process called remapping (*Muller and Kubie, 1987*; *Dragoi et al., 2003*; *Leutgeb et al., 2005*; *Dragoi and Tonegawa, 2011*, *2013*). Consistent with the remapping, the order in which the place cells fired in the familiar track (place cell sequences) was not correlated with their order of firing in the novel arm (contiguous condition, mean $R_{CT}^2$ = 0.03, mean $R_{KO}^2$ = 0.12, p>0.05, both genotypes) or novel track (disjunct condition, mean $R_{CT}^2$ = 0.11, mean $R_{KO}^2$ = 0.12, p>0.05, both genotypes), whereas it was similar to the order in which they later fired in the familiar arm of the L-shape track (contiguous condition, $R_{CT}^2$ = 0.6, $R_{KO}^2$ = 0.5, p<0.05, both genotypes).

## Prior experience- and CA3 NMDAR-dependent changes in spatial-temporal neuronal sequence correlations

For each run session and for each direction of movement, place cells were ordered according to the location of their peak firing (>1 Hz) on the corresponding track/arm, resulting in two place cell sequence templates for each condition and session, one for each direction (*Dragoi and Tonegawa, 2011*). Spiking events (*Foster and Wilson, 2006*; *Diba and Buzsaki, 2007*; *Dragoi and Tonegawa, 2011*, *2013*) were detected for each sleep/rest session in the sleep/rest box as increases in multiunit activity (at least five of the place cells active during the corresponding run session) that were preceded and followed by >100 ms of silence. For each spiking event, a rank-order correlation between the place cell sequence template and the temporal sequence of cell firing during the event was calculated for each direction of runs and for each session and experimental condition in both CT and KO mice (*Figure 3A–F*). The event was considered significant if the correlation of its firing sequence with the corresponding place cell sequence template exceeded the 97.5th percentile of a distribution of correlations when the order of the place cells in the novel track template or novel arm template was shuffled randomly 100 times (i.e., p<0.025). For each direction of run, preplay (i.e., pre-Run play) refers to an event's temporal sequence during pre-Run sleep/rest that has a significant correlation with the spatial cell sequence of the subsequent run session. Likewise, post-Run play or replay refers to an event's temporal firing sequence during post-Run sleep/rest that has a significant correlation with the spatial cell sequence of the preceding run session. Events significant for both running directions were assigned only to the direction with the higher absolute correlation value.

For both genotypes and for all experimental conditions and sessions, the absolute correlation values between events occurring during the pre-Run or post-Run sleep/rest session and the original novel track/arm templates (*Figure 4A–H*) were significantly higher than the correlation values obtained using corresponding shuffled templates (*Figure 4*, left panels, ranksum test, see 'Materials and methods'; all mice passed individual significance for preplay and replay across conditions, ranksum or binomial probability tests). At the level of individual animals, with the exception of one CT mouse

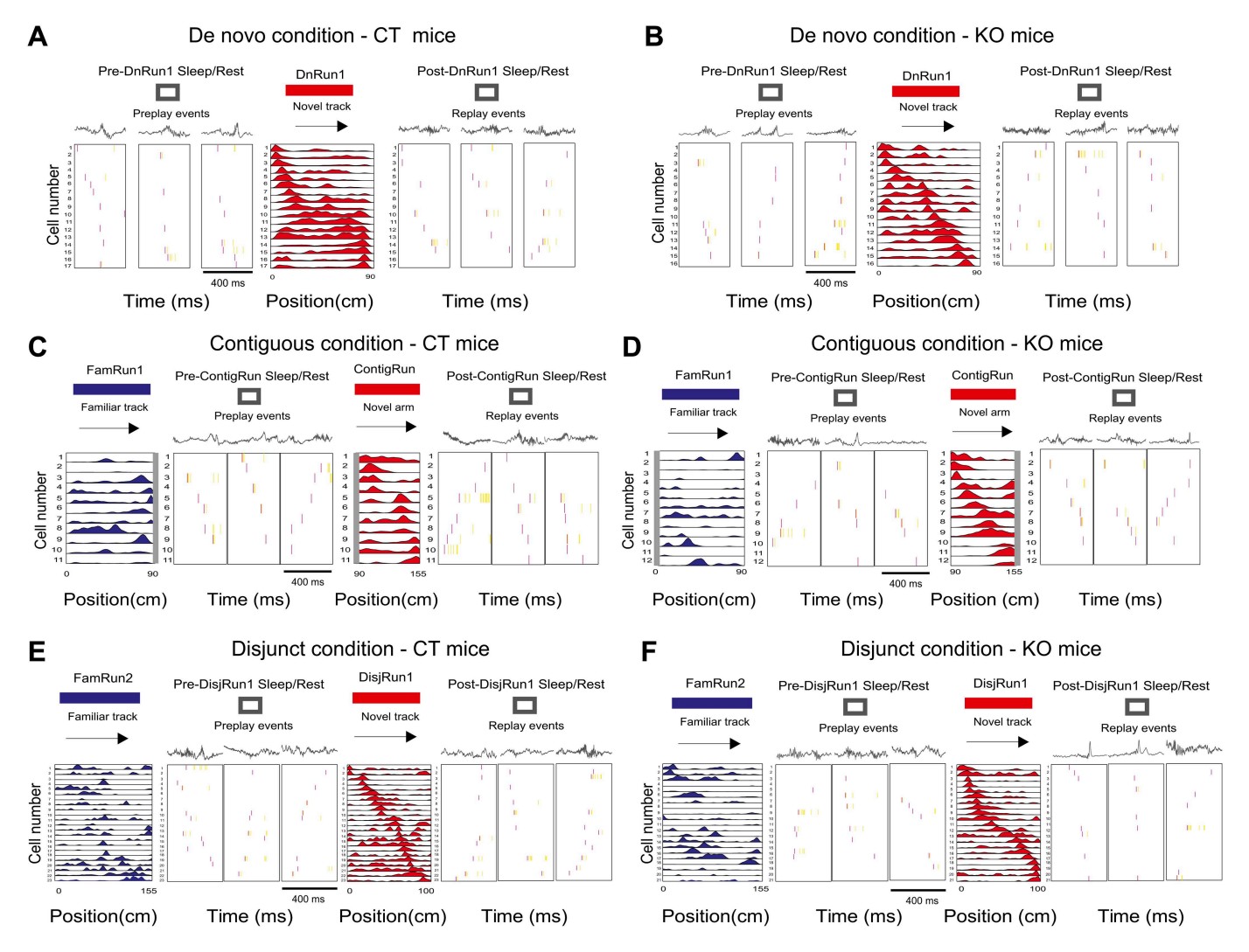

**Figure 3**. Display of preplay and replay sequences during sleep/rest in naïve and experienced mice in all experimental conditions. (**A** and **B**) Examples of preplay spiking events during the sleep/rest session in the sleep/rest box preceding the first run on the novel track during De novo condition (first three boxes from the left), the corresponding place cell sequence during run (fourth box), and replay events during the sleep/rest session in the sleep/rest box following the run (last three boxes), in one control mouse, CT2 (**A**) and one CA3 NMDAR KO mouse, KO2 (**B**). Arrows indicate the order of the place cell sequence. Corresponding local field potential recordings are shown above the spiking events. (**C** and **D**) Examples of preplay spiking events during the sleep/rest session in the sleep/rest box (second to fourth boxes from the left) following a run session on the familiar track (leftmost box, place cells in blue) and preceding the run session on the novel portion of the L-shape track under the contiguous condition (fifth box, place cell sequence in red), and examples of replay events during the post-run sleep/rest (last three boxes), from one control mouse, CT2 (**C**) and one CA3 NMDAR KO mouse, KO2 (**D**). (**E** and **F**) Examples of preplay, place cells, and replay sequences before, during, and after run on an isolated novel linear track (place cells in red) in the disjunct condition that all followed run on a familiar track (place cells in blue), in one CT (**E**) and one KO (**F**) mouse. Boxes are assigned to experimental sessions like in (**E** and **F**). For all subplots, spikes in red during spiking events represent the first spike for each participating cell; all the other spikes are in yellow. The place cell sequence template panels shown in (**C**) for mouse CT2 (left, blue and right, red) are reproduced from *Figure 1Ea,Ec*; *Dragoi and Tonegawa (2011)*, Nature; Nature Publishing Group has granted permission to reproduce these images under the terms of the Creative Commons Attribution 3.0 Unported License.

(CT2, preplay > replay, $p<10^{-4}$, ranksum test), the absolute values of the correlations of the significant events with the spatial templates of the very first run session on the novel track (i.e., DnRun1) were similar for the spiking events occurring during the pre-DnRun1 sleep/rest session (i.e., when preplay occurs) compared to those occurring during the post-DnRun1 session (i.e., when replay occurs), both in CT and KO mice (*Figure 5Aa*, $p>0.05$, ranksum test). The proportions of significant events out of all

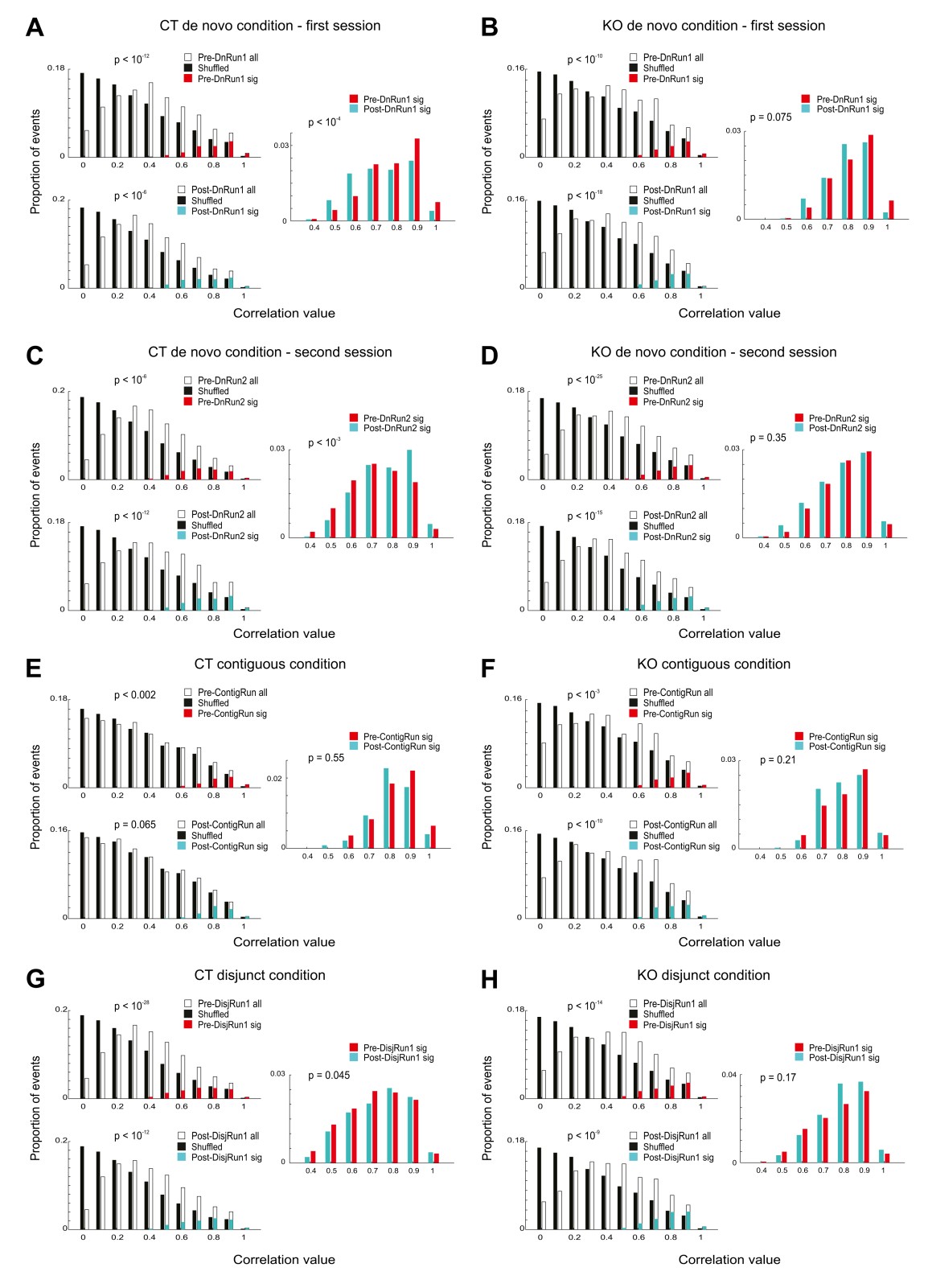

**Figure 4**. Significance of preplay and replay events across experimental conditions and genotypes. (**A–H**) Distribution of absolute values of spiking event-place cell sequence correlations for all events occurring during the pre-Run sleep/rest (open bars, Pre-Run all; solid black bars, corresponding shuffle correlations), all events during post-Run sleep/rest (open bars, Post-Run all; solid black bars, corresponding shuffle correlations), significant

*Figure 4. Continued on next page*

Figure 4. Continued

pre-Run events (red bars, Pre-Run sig), and significant post-Run events (blue bars, Post-Run sig) from all CT and all KO mice corresponding to the sessions and conditions described in *Figure 1A*. The conditions, sessions, and the genotype are specified above each subplot (**A–H**). p values reflect differences between corresponding distributions using ranksum test. Data in **A** and **C** are from mice CT1-4; data in **B** and **D** are from mice KO1-3; data in **E** are from mice CT2-4; data in **F** are from mice KO1-4. Data in **G** and **H** are from three CT and three KO mice.

events were also similar in the pre-DnRun1 vs post-DnRun1 sleep/rest for each individual mouse and genotype (p>0.05 for each individual animal, Z-test for two proportions, *Figure 5Ba*; preplay vs replay: 9/132 vs 9/148 in CT1, 237/2255 vs 438/4456 in CT2, 2/10 vs 5/45 in CT3, 7/136 vs 5/71 in CT4, 2/48 vs 4/74 in KO1, 104/1310 vs 112/1496 in KO2, and 80/1146 vs 141/1824 in KO3). Surprisingly, when the absolute correlation values of the significant events were pulled together from all animals, they were higher for the spiking events occurring during the pre-DnRun1 sleep/rest session (preplay) compared to those occurring during the post-DnRun1 session (replay) in CT but not KO mice (*Figure 4A–B*, right panels, *Figure 5Ca*, empty bars; pre-DnRun1 compared to post-DnRun1, 0.87 ± 0.008 (std. 0.12) compared to 0.80 ± 0.007 (std. 0.13), p<$10^{-4}$ in CT, *Figure 4A*, right panel, *Figure 5Ca*, empty bars, left; and 0.89 ± 0.008 (std. 0.1) compared to 0.88 ± 0.007 (std. 0.1), p>0.075 in KO, *Figure 4B*, right panel, *Figure 5Ca*, empty bars, right, ranksum test).

Given the relative instability of the place fields in the very first run session in naïve animals (*Figure 1E*), we asked whether the initial and the later spatial templates display different spatial-temporal correlation values with the temporal sequences during the preceding sleep. In order to address this question, we constructed spatial templates from the activity of place cells in the first four laps of run (early templates) and the last four laps (late templates) and correlated them with the temporal sequences recorded during the previous sleep (i.e., early and late preplay). We found that the population of early template correlations was not different from the population of late ones in both CT and KO mice (*Figure 5—figure supplement 1*). However, both early and late spatial-temporal correlations were higher in the KO mice compared with the CT ones (*Figure 5—figure supplement 1*). These results indicate that the relative instability of the place fields during DnRun1 session is associated with a process of relatively mild 'editing' of the early template, rather than with a process of dramatic 'remapping' into a new chart. The relatively high correlation between early and late place fields in the de novo condition (0.4–0.5) is consistent with this scenario.

Following a second exposure to the novel track, with the exception of one CT mouse (CT2, preplay<replay, p<0.006), the significant correlations of spiking events with the DnRun2 templates were similar during post-DnRun2 sleep/rest compared to the pre-DnRun2 sleep/rest in CT and KO mice (*Figure 5Ab*, p>0.05, ranksum test). Moreover, the proportions of significant events were also similar before and after the DnRun2 run experience in all mice (p>0.05, Z-test for two proportions, *Figure 5Bb*; proportions preplay vs replay: 15/164 vs 5/69 in CT1, 465/4567 vs 196/1753 in CT2, 21/160 vs 196/1753 in CT3, 3/70 vs 8/131 in CT4, 3/74 vs 8/66 in KO1, 186/1942 vs 261/2578 in KO2, and 223/2504 vs 158/1807 in KO3). When absolute correlation values were pulled together across animals, the pre-Run vs post-Run play relationship reversed in the CT but not in the KO animals (*Figure 4C–D*, right panels, *Figure 5Cb*, empty bars). The significant correlations of spiking events with the DnRun2 templates were stronger during post-DnRun2 sleep/rest compared to the pre-DnRun2 sleep/rest in CT animals (pre-DnRun2 compared to post-DnRun2, 0.77 ± 0.006 (std. 0.13) compared to 0.83 ± 0.008 (std. 0.13), p<$10^{-3}$, ranksum test, *Figure 4C*, right panel, *Figure 5Cb*, empty bars, left); however, they were similar in the KO mice (*Figure 4D*, right panel, *Figure 5Cb*, empty bars, right; pre-DnRun2 compared to post-DnRun2, 0.85 ± 0.006 (std. 0.11) compared to 0.85 ± 0.006 (std. 0.12), p>0.35).

In the contiguous condition, both at the individual animal level (*Figure 5Ac*) and at the group level the correlations with the ContigRun templates in the post-ContigRun (i.e., replay) were similar to the ones in the pre-ContigRun sleep/rest (i.e., preplay) for both genotypes (group level; *Figure 4E*, right panel, *Figure 5Cc*, empty bars, left, pre-ContigRun compared to post-ContigRun, 0.90 ± 0.01 (std. 0.1) compared to 0.87 ± 0.009 (std. 0.09), p>0.55 in CT mice; *Figure 4F*, right panel, *Figure 5Cc*, empty bars, right, pre-ContigRun compared to post-ContigRun, 0.89 ± 0.01 (std. 0.1) vs 0.86 ± 0.007 (std. 0.09), p>0.21 in KO mice, ranksum test). The proportions of significant events out of all events were also similar before and after the ContigRun experience (*Figure 5Bc*, p>0.05, Z-test for two proportions; preplay vs replay incidence: 59/1013 vs 83/1601 in CT2, 3/23 vs 21/180 in CT3, 2/51 vs 23/459 in CT4, 28/381 vs 74/903 in KO2, 55/788 vs 114/1422 in KO3, and 7/123 vs 23/418 in KO4).

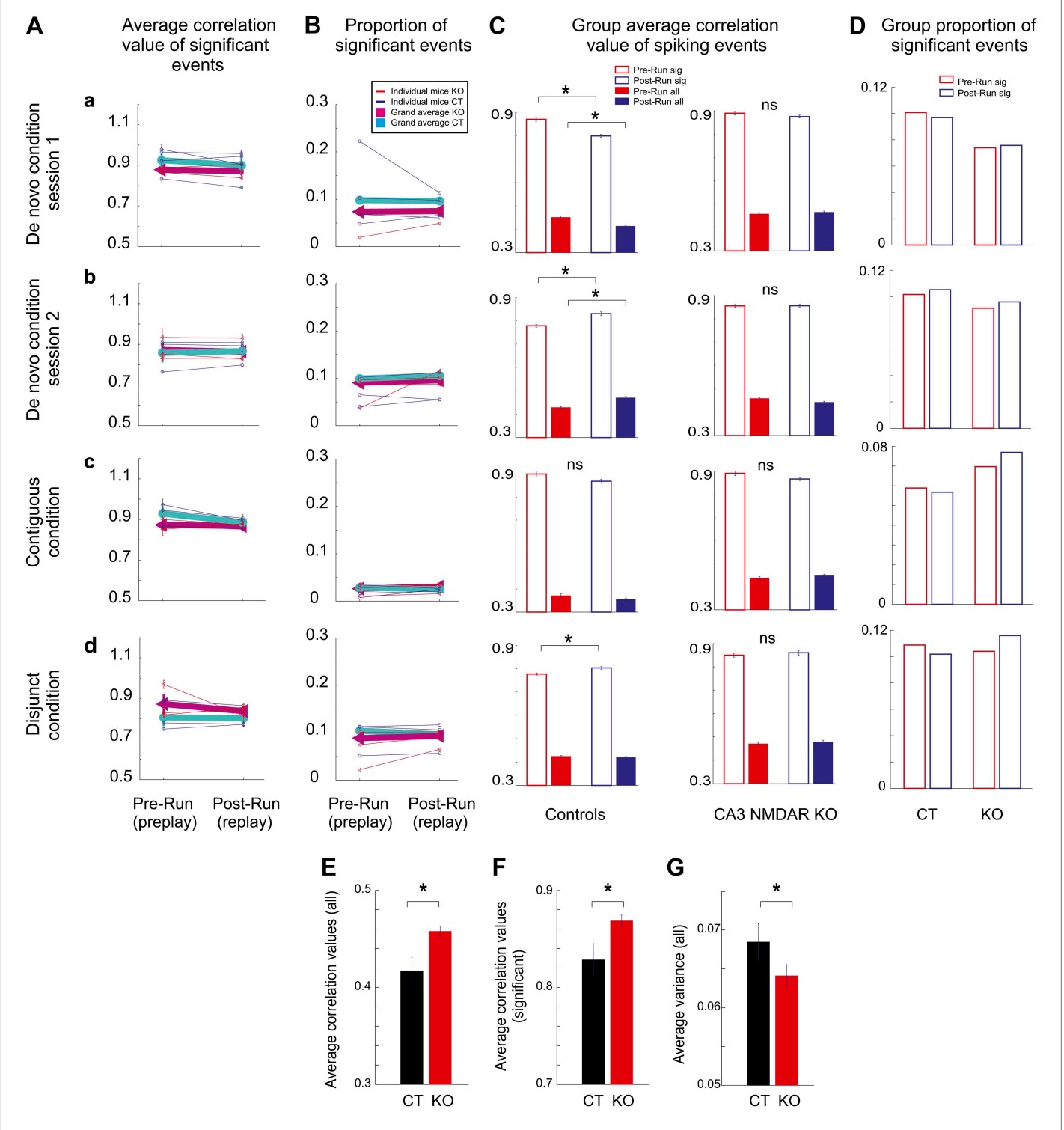

**Figure 5**. Comparison of hippocampal temporal sequence activity during sleep/rest across multiple novel spatial experiences in individual mice. (**A**) Average correlation values for preplay and replay in individual animals and group data. Small letters, (**a–d**), represent different experimental conditions and sessions for **A-D**. Thin lines represent individual animals; thick lines represent grand averages across all animals by condition/session and genotype for **A** and **B**. (**B**) Incidence of significant temporal sequences (preplay and replay) in individual animals and group data. Group comparison between preplay and replay during sleep/rest in naïve and experienced control and CA3 NMDAR KO mice. (**C**) Group average absolute correlation values of spiking event-place cell sequence correlations during pre-Run (red bars) and post-Run (blue bars) sleep/rest sessions. Left: correlations in control animals; right: correlations
*Figure 5. Continued on next page*

*Figure 5. Continued*

in KO animals. Solid bars: correlations calculated during all spiking events; empty bars: correlations calculated during significant events. Error bars represent SEM. Stars mark significant differences. (**D**) Group proportion of significant preplay and replay events across conditions, sessions, and genotypes. (**E–G**) Flexibility/rigidity of spatial-temporal sequences between sleep/rest and run across genotypes. (**E–F**) Averages of correlations between firing sequences during spiking events in sleep/rest (preplay and replay) and place cell sequences (all events, **E**; significant events, **F**). (**G**) Comparison of variance in the distribution of all correlation values from **E**. Error bars are SEM. Stars denote statistical significance.

The following figure supplements are available for figure 5:

**Figure supplement 1**. Comparison between preplay of place cell sequences computed from the activity in the early vs late parts of the de novo DnRun1 session in CT and KO mice.

**Figure supplement 2**. Similar features of preplay and replay during sleep/rest across multiple novel spatial experiences in individual mice.

**Figure supplement 3**. Histograms depicting co-occurrence of ripple oscillations and preplay/replay events across session, conditions, and genotypes.

**Figure supplement 4**. Immunohistochemistry for NR1 subunit of the NMDA receptor demonstrating the absence of the NMDA receptors specifically in the CA3 area in the KO mice (left) and its preservation in all areas of the hippocampus in the floxed NR1 CT mice.

In the disjunct condition, at the level of individual animals, the absolute correlation values of significant events (*Figure 5Ad*) and their incidence were similar before and after the DisjRun1 experience (*Figure 5Bd*, p>0.05 for all mice, Z-test for two proportions; preplay vs replay incidence: 541/4822 vs 265/2503 in CT2, 62/824 vs 74/1024 in CT3, 313/2769 vs 164/1410 in CT4, 5/102 vs 3/18 in KO1, 185/1671 vs 33/311 in KO3, and 41/446 vs 103/869 in KO4). At the genotype group level, the novel experience (DisjRun1) resulted in slightly increased correlations in the post-DisjRun1 vs pre-DisjRun1 sleep/rest in CT (0.80 ± 0.006 (std. 0.14) vs 0.77 ± 0.005 (std. 0.14), p<0.05, ranksum test, *Figure 4G*, right panel, *Figure 5Cd*, empty bars, left), but not KO animals (0.86 ± 0.01 (std. 0.12) vs 0.85 ± 0.009 (std. 0.11), p>0.17, *Figure 4H*, right panel, *Figure 5Cd*, empty bars, right).

Across all experimental conditions and sessions and for both genotypes there were no overall changes in the number of cells active per significant event (preplay vs replay, *Figure 5—figure supplement 2A*, p>0.05, ranksum test), in the duration of significant events (*Figure 5—figure supplement 2B*, p>0.05, ranksum test), and in the extent of the linear track being represented during the significant events (*Figure 5—figure supplement 2C*, p>0.05, ranksum test). For both genotypes and in all pre- and post-run sleep/rest sessions, the time of occurrence of significant preplay and replay events correlated with the time of occurrence of high-frequency oscillation ripples in the CA1 (*Figure 5—figure supplement 3*).

Interestingly, the experience- and CA3 NMDAR-dependent changes in replay over preplay described at the animal group level across sessions in the de novo condition was also found when the correlation values between place field templates and *all* spiking events (i.e., not only the significant ones) were considered (*Figure 5Ca,b*, solid bars). The correlation values of all spiking events during pre-DnRun1 sleep/rest exceeded the correlation values of all spiking events recorded during post-DnRun1 sleep/rest in CT but not KO mice (p<10$^{-5}$ in CT, p>0.5 in KO, *Figure 5Ca*, solid bars). Preplay correlations were not different across genotypes (p>0.9, ranksum test), but replay correlations were higher in the KO than CT mice (p<10$^{-8}$, ranksum test). Moreover, after a second experience on the novel track (*Figure 5Cb*, solid bars), the correlation values of all post-Run (i.e., replay) events became higher than the corresponding values of all pre-Run events in CT (p<10$^{-5}$) but not KO animals (p>0.5). Despite the changes in correlation values from preplay to replay in CT mice, the proportions of significant events out of all of the detected events in all animals were similar in the pre-Run and post-Run sessions under all experimental conditions for both genotypes (*Figure 5D*, p>0.05, Z-test for two proportions). The incidences of spiking events were similar in the sleep/rest sessions preceding and following the novel run experiences across conditions in both CT (p>0.08, paired *t* test) and KO mice (p>0.9).

More importantly, comparison of averages of absolute preplay and replay correlations over all experimental conditions and sessions performed altogether between the two genotypes (eight sessions/genotype, paired by session type between genotypes) revealed that correlation values were significantly higher (p<0.006 for the significant correlations; p<0.009 for all the correlations; paired *t* test) and their variance was significantly lower (p<0.007, paired *t* test) in KO vs CT mice (*Figure 5E–G*). This finding is consistent with the overall reduced experience-dependent changes in the correlation

values in KO (*Figures 4B,D,F,H and 5C*) and indicates a reduced flexibility (increased rigidity) of the hippocampal network in the absence of NMDAR-dependent activity in the CA3 region. Moreover, the overall spatial extent (i.e., proportion) of the linear track represented by the significant spiking events during sleep/rest was higher in the CT vs KO mice (0.94 ± 0.01 compared to 0.89 ± 0.01 of the track length, p<0.027, paired *t* test), indicating a role for the NMDAR-dependent activity in the CA3 area in temporally linking together chunks of spatial sequences (*Dragoi and Buzsaki, 2006*; *Gupta et al., 2012*). These differences in correlations between genotypes were not due to overall differences in the duration of the significant spiking events (KO compared to CT across sessions, 0.33 ± 0.02 s compared to 0.33 ± 0.02 s, p=0.82, paired *t* test), or in the number of cells participating in these events (KO compared to CT, 6.1 ± 0.2 compared to 6.5 ± 0.2, p=0.65, paired *t* test) across genotypes.

## Behavioral performance on T-maze delayed alternation tasks

We tested the behavioral relevance of the observed differences in neuronal dynamics by measuring the animal performance on a hippocampal-dependent learning task, the delayed T-maze alternation (*Ainge et al., 2007*). In this task, the animals had to retrieve food rewards placed at the end of left vs right arms of the T-maze in alternate trials. The animals self-initiated the trials, moved toward the reward site and returned on the same path to re-initiate the next trial. Delays between trials were counted as the time between returning from the end of the left/right arms and self-initiation of the next trial (>10 s). Both groups of animals reached the criterion level of performance of 70% correct choices during the 10-day training (*Figure 6A*). Analysis of the data from both genotypes during the 10-day training using a balanced two-way ANOVA test (five blocks of 2 consecutive days) revealed a significant effect of the training day (F = 11.83, p=0.006) and genotype (F = 10.5, p=0.001) and a marginal duration (day) × genotype interaction (F = 3.24, p=0.057). Consistent with the difference in the neuronal dynamics across genotypes during the de novo condition (*Figure 2A–D*, left), the KO subjects required additional sessions to reach the 70% criterion for learning to alternate (9 vs 7 days, KO vs CT, *Figure 6A*; days 7–8 block, CT vs KO, 78.1% vs 53.7% correct, p=0.006, ranksum test)

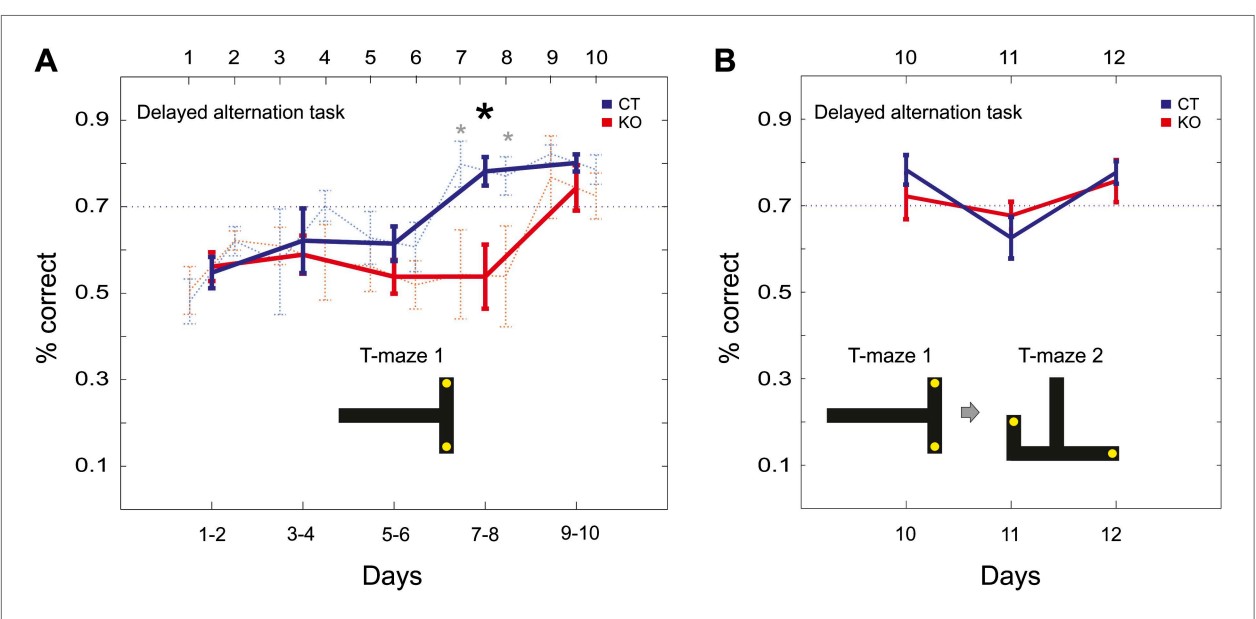

**Figure 6**. Behavioral performance of CT and KO mice in a T-maze alternation task. (**A**) Performance in the alternation task over the 10 days of training on the first T-maze configuration (T-maze 1; 70% is the performance criterion for learning). Thin lines, learning curves expressed as averages of all subjects per group in the 10 days of training. Bold lines, learning curves expressed as averages of all subjects per group in blocks of two consecutive days of training. (**B**) Performance in the alternation task on exposure to a second T-maze configuration (T-maze 2). Note that for both genotypes, the performance in T-maze 2 on day 11 dropped below the 70% correct threshold. In the CT group, the performance in T-maze 2 on day 11 was significantly lower compared to the performance in T-maze 1 on day 10 (p<0.05, paired t-test). This indicates the animals detected (and subsequently learned) the novel spatial configuration of T-maze 2. Error bars are SEM. Stars mark significant differences between groups. Insets: cartoons of the T-maze configurations in thick black lines; yellow dots, location of food reward at the ends of the choice arms.

indicating the involvement of the intrinsic hippocampal circuitry in the acquisition of this task. Moreover, consistent with the facilitation of neuronal dynamics on the novel arm by prior experience on the familiar track during the contiguous condition (*Figure 2A–D*, middle), this learning experience greatly accelerated (*Tse et al., 2007*) the acquisition of a similar alternation task in a second, novel T-maze configuration, when both groups of animals performed above threshold by the second day of re-training (*Figure 6B*, day 10 vs 12, 78.2% vs 77.6% correct, p=0.9 for CT, and 72.2% vs 75.7% correct, p=0.6 for KO, ranksum test). Together, these results establish a correlation between the neuronal activity and the behavioral performances across genotypes and experimental conditions and suggest that the dynamics of cell assembly organization are part of the underlying mechanisms of spatial learning.

## Discussion

### De novo and related novelty learning

Previous studies on spatial representation in the hippocampus either investigated the process of spatial encoding by evaluating the dynamics of place field formation and neuronal correlation during exploration (*Hill, 1978*; *Wilson and McNaughton, 1993*; *Nakazawa et al., 2003*; *Frank et al., 2004*; *Cacucci et al., 2007*; *Brun et al., 2008*) or, separately, described the phenomenon of temporal sequence replay during awake resting (*Foster and Wilson, 2006*; *Diba and Buzsaki, 2007*; *Davidson et al., 2009*; *Karlsson and Frank, 2009*) or post-Run sleep (*Nadasdy et al., 1999*; *Lee and Wilson, 2002*; *Ji and Wilson, 2007*; *Karlsson and Frank, 2009*). Importantly, all of these studies used experienced animals which were either pre-trained on similar environments or were simply re-exposed to the familiar ones, and whose prior neuronal activity as naïve animals in previous environments was not documented. Overall, the expression of spatially tuned novel place fields in experienced animals as reported in some of the previous studies (*Hill, 1978*; *Nakazawa et al., 2003*) is consistent with our findings in experienced CT mice (contiguous and disjunct conditions), though a faster timescale analysis might reveal additional dynamics in spatial tuning (*Frank et al., 2004*).

Our approach is to study the process of internal development of novel spatial representations as a dynamic whole by comparing and correlating the activity of ensembles of neurons during the sleep/rest period prior to first time exploration of a linear track with the one during the exploration, and both of these activities with the one during the post-Run sleep/rest session, in naïve and experienced animals, in the presence and absence of CA3 NMDARs.

This approach allowed us to identify and compare three distinct forms of novelty encoding as revealed by prior experience and CA3 NMDAR KO. In the contiguous and disjunct condition paradigms, although the spatial location and orientation of the novel linear track are as new to the animals as in the de novo condition paradigm, both the geometry (i.e., linear tracks) and the behavioral experience (i.e., repetitive runs for food rewards) are common. These parameters may have already been internalized prior to the novel run session, which may have diminished the dynamic interplay between the internal and external drives and facilitated the formation of more stable spatial representations on the novel arm/track. More importantly, the repeated access to the familiar arm in conjunction with the exploration of the novel arm in the contiguous condition accelerated the recruitment and stabilization of neuronal firing sequences independent of CA3 NMDARs; this likely happened through complementary, non-CA3 NMDAR-dependent plasticity or through NMDAR-dependent activity in other brain regions (*Kentros et al., 1998*). The CA3 NMDAR independence in the contiguous condition did not solely result from increased experience with linear tracks, as exploration of isolated novel tracks by the even more experienced animals during the disjunct condition did require, transiently, these receptors for the rapid formation of stable, tuned place cell sequences. In the disjunct condition, since exposure to the novel track occurred in the same general spatial environment, modules of place cells that remapped together, possibly controlled by common external stimuli (*Lee et al., 2004*), were regrouped in a CA3 NMDAR-dependent manner to rapidly form the new representation. We hypothesize that this regrouping process observed in the disjunct condition makes the need for CA3 NMDARs only transient and thereby facilitates the formation of a new spatial representation based on the prior experience. A more drastic change in the external stimuli in the absence of prior animal training (like in the de novo condition where animals were shifted for the first time from the sleep/rest box to the linear maze) may lead to a more complete CA3 NMDAR-dependent recruitment and stabilization of neuronal ensemble activity, a slower formation of new spatial representations, and a slower learning.

## Rapid dynamics of cell assembly organization in the CA1 during exposure to novel environments and the role of CA3 NMDARs

Our data have shown that a novel representation of a first-time experience on linear tracks (DnRun1) is formed in the CA1 area primarily on the framework of the preconfigured hippocampal network (preplay), which is modified, in part, during the experience and is rapidly stabilized primarily via CA3 NMDAR-dependent activity (*Figure 7*). The changes in place cell activity occur without affecting the general stability of the hippocampal network, which indicates a homeostatic regulation of its temporal sequence activity in response to novelty: ~10% of the temporal sequences reflect the novel experience during the subsequent sleep/rest period, a proportion that is not different from the proportion of corresponding preplay events. Subsequent exposures to the same track (DnRun2) and additional contiguous (ContigRun) or isolated novel tracks (DisjRun1) resulted in the expression of similar proportions of correlated temporal sequences before and after the corresponding novel spatial experiences, both in control and CA3 NMDAR KO mice.

We propose that the overall robust homeostasis of the hippocampal network seen at the temporal sequence level and expressed during offline states of sleep and rest reflects the default sequential cell assembly architecture of the hippocampal network shaped by the multiple unaccounted experiences the animal has had in the past. Our proposal is consistent with previous studies reporting that temporal sequences emitted during onsite resting periods do not specifically reflect the recent spatial experience of the animal (*Karlsson and Frank, 2009*; *Gupta et al., 2010*; *Dragoi and Tonegawa, 2011*; *Pfeiffer and Foster, 2013*), but rather reflect multiple related spatial experiences the animal had experienced or will experience in the near future in that particular environment (*Dragoi and Tonegawa, 2011*, *2013*).

The overall stability of the hippocampal network during sleep/rest epochs on both sides of the novel spatial experiences does not mean that the novel experiences did not induce more discrete plastic changes which are apparent at the individual cell level in the network (*Dragoi et al., 2003*). More importantly, in the absence of CA3 NMDAR-dependent activity, the CA1 temporal firing sequences appear more rigid and their correlations with the place cell sequences are less modulated by the recent experiences compared to control animals. Overall, in the absence of CA3 NMDARs, the event correlations with future place cell sequences exhibit lower variance and higher values than in the presence of CA3 NMDARs. These results indicate that the organization of cellular assemblies in the CA1 area is influenced by the NMDAR-dependent activity in the upstream CA3 area. In the absence of this type of activity/plasticity, the CA1 cellular assemblies are less affected by the animals' novel

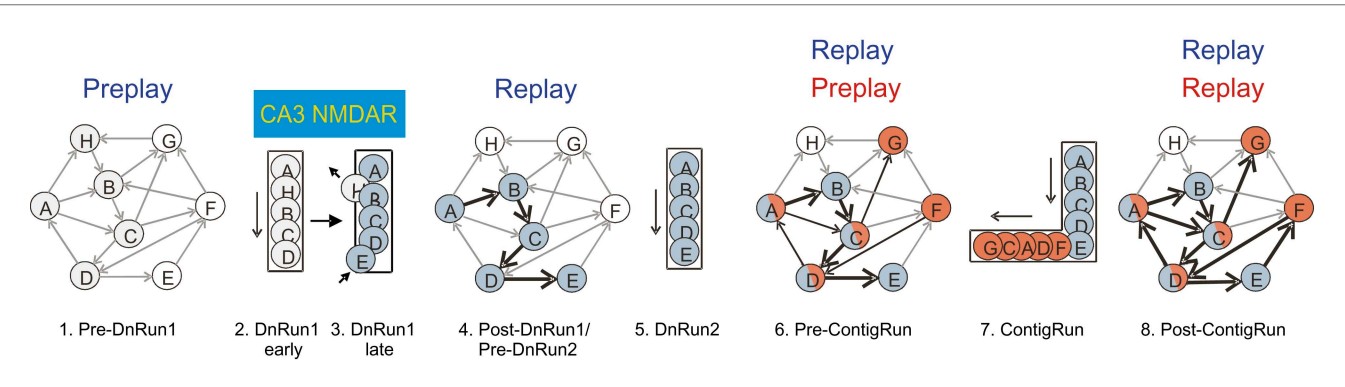

**Figure 7**. Cartoon model of the internal representation of novel experiences based on the organization of neurons in cellular assemblies. 'Hexagonal' panels (panels 1, 4, 6, 8), network of sequentially activated neurons (cell assemblies) during sleep/rest under different experimental conditions: pre-DnRun1 sleep/rest (panel 1), post-DnRun1 sleep/rest (panel 4), pre-ContigRun sleep/rest (panel 6), and post-ContigRun sleep/rest (panel 8). Arrows indicate potential (thin) or actual (bold) temporal order of activation during Run in CA1, *not* anatomical connectivity. All arrows during sleep/rest indicate the temporal order of cell activation during sleep/rest. Black bold arrows during sleep/rest emphasize temporal replay. Upper case letters: corresponding individual cells/assemblies. Colors: sequential cell assemblies co-active on a given linear track. Cells A, C, and D are active on both the familiar and novel arms. White circles: cells with no place field during the corresponding run session. 'Linear' panels (panels 2, 3, 5, 7), sequences of place cells during different sessions of run on linear tracks: DnRun1 (early, panel 2 and late, panel 3, of the run session), DnRun2 (panel 5), and ContigRun (panel 7). Letters, colors, and order of activation correspond to the ones during sleep/rest sessions. Long thin arrows next to the panel represent the direction of the animal's movement during run.

spatial experiences and maintain an increased correlation across different brain states (i.e., across sharp-wave/ripples during sleep and theta during run), behaviors, and experiences. This could explain why the mutant animals exhibit deficits in one trial learning which involves rapid plastic changes in hippocampal cellular assemblies (*Nakazawa et al., 2003*).

A hallmark of neurons' organization in cellular assemblies is their onsite coordinated activation across similar animal behaviors such as running laps (*Wills et al., 2005*; *Dragoi and Buzsaki, 2006*). Here we show that upon familiarization with a novel linear track, the lap-by-lap correlations between place cell pairs increase rapidly during the exploratory session in control animals but remain low for several exploratory sessions in the absence of CA3 NMDARs. These results suggest that NMDAR-dependent activity within the CA3 area of the hippocampus is involved in the rapid organization and linking of CA1 place cells in cellular assemblies during the encoding of a first-time and of subsequent isolated novel experiences on linear tracks, but is not necessary for the general expression of temporal sequences of place cells in the CA1 in the form of preplay and replay.

Our results have relevance for theories of learning and memory consolidation. Quite often, a single exposure to absolutely novel experiences does not lead to a lasting memory of the experience, whereas repeated exposures to the same experience or prior knowledge with similar kind of experiences result in more rapid learning and memory consolidation (*Tse et al., 2007*). We propose that the increased map stability, tuning, and cell assembly co-variation we find in repeatedly exposed and in experienced, but not in naïve animals underlie the rapid consolidation of episodic memories of repeated or related new experiences, but not of entirely novel ones. The latter may instead require repeated exposure to engage hippocampal replay-dependent mechanisms for memory consolidation.

### Prior knowledge and schema-based learning

The intact hippocampus is essential for encoding and rapid consolidation of memory (*Scoville and Milner, 1957*; *Squire, 1992*; *Eichenbaum et al., 1999*) and for associative-novelty detection (*Kumaran and Maguire, 2007*). The role of the hippocampus and of NMDA receptors in learning and memory consolidation is manifested primarily in experimentally naïve animals and diminishes with experience (*Bannerman et al., 1995*; *Otnaess et al., 1999*), when new information is presumably rapidly integrated into pre-existing neocortical frameworks of knowledge, or schemas (*Morris, 2006*; *Tse et al., 2007*; *McKenzie and Eichenbaum, 2011*). The neural substrates of such learning and memory processes are believed to be the formation of stable, finely tuned cellular assemblies (*Hebb, 1949*) across the neocortex and the hippocampus (*Tse et al., 2011*).

We propose that, in addition to changes in the activity of individual CA1 neurons, their ability to rapidly and flexibly organize in stable cellular assemblies underlies the process of learning and memory. Familiarization with the spatial environment and the behavioral task are associated with the formation of cortical mental schemas (*Tse et al., 2007*; *McKenzie et al., 2013*) that rely on the stability of neocortical-hippocampal cellular assemblies. These function like strong neural attractors (*Tsodyks, 1999*; *Wills et al., 2005*) that will integrate future neuronal representations. The group of CA1 place cell pairs that maintained high levels of lap-by-lap co-variation and preserved the relative distance between their place fields across different environments independent of CA3 NMDARs (stable assemblies, disjunct condition; contiguous condition) may represent one neuronal mechanism underlying schema-based accelerated learning.

Previously, lesion experiments have argued for a transient, but necessary role of the intact hippocampus in the assimilation and consolidation of new information into schemas (*Tse et al., 2007*). Whereas not speaking for the whole hippocampus, our data indicate that CA3 NMDAR-dependent activity and synaptic plasticity are not necessary for either the rapid assimilation of new contiguous locations into a previously established spatial representation or for learning of a related alternation task. Instead, this type of plasticity is necessary for the rapid formation of new CA1 cell assemblies in the hippocampus of experimentally naïve animals and the development of a new schema associated with the first-time learning of a hippocampal-dependent alternation task (*Figure 7*). The existence of a mechanistic dichotomy between these different forms of learning may help explain why hippocampal dysfunction results in anterograde amnesia (*Scoville and Milner, 1957*) while recollection of old, schema-based memories (*Winocur et al., 2005*) is preserved.

## Materials and methods

A total of 20 adult male mice with ages between 18–22 weeks were employed in this study. Half of them (n = 10) had the NMDA receptor subunit NR1 deleted specifically in the CA3 area of the hippocampus

(*Nakazawa et al., 2002*) leading to a complete loss of NMDA receptors in the CA3 pyramidal cells (the KO group). The other half (the CT group) was represented by the floxed-NR1 mice (*Tsien et al., 1996*). The CT mice were littermates of the CA3 NMDAR KO mice; both CT and KO mice had been back-crossed to C57BL/6 mice. The knock-out of the NR1 subunit was achieved using the Cre/loxP recombination system (*Tsien et al., 1996*) and was present in homozygous floxed, Cre-positive mice starting with the age of 15 weeks (*Nakazawa et al., 2002*).

In vivo electrophysiological recordings were collected from eight mice (four CT and four KO) implanted with recording electrodes on the right side of the hippocampus. After the completion of the experiments, the brains of all mice were perfused and fixed. The right side of the brain was sectioned and stained using Nuclear fast red or Cresyl violet for electrode track reconstruction; the left side of the brain was processed for immunohistochemistry for the NR1 subunit to confirm the lack/presence of NMDA receptors in the CA3 area of each KO/CT participating mouse (*Figure 5—figure supplement 4*). The remaining 12 mice (six CT and six KO) were subjected to the T-maze delayed alternation task. All mice were re-genotyped post experiments to confirm their inclusion in the CT and KO group.

## Experimental design of electrophysiological experiments

All animals were implanted under *Avertin* anesthesia with six independently movable tetrodes aimed at the CA1 area of the right hippocampus (1.5–2 mm posterior to bregma and 1–2 mm lateral to the midline). The reference electrode was implanted posterior to lambda over the cerebellum. During the following week of recovery, the electrodes were advanced daily while animals rested in a small sleep/resting box (12 × 20 × 35 [hr] cm) having opaque walls. The animal position was monitored via two infrared diodes attached to the headstage.

The experimental apparatus consisted of a 90 × 65 cm rectangular, walled linear track maze. All tracks were 4 cm wide at the bottom and 8–9 cm at the top; all linear track walls were translucent, 10 cm high, with opaque, uniform color barriers. Recording sessions (*Table 1*) were conducted while the animals explored for chocolate sprinkle rewards placed always at the ends of the corresponding linear tracks (one sprinkle at each end of the track on each lap).

Under the de novo condition, the neuronal activity was recorded in naïve animals (four CT mice, CT1–4, and four KO mice, mice KO1–4) during the sleep/rest session in the sleep/rest box (pre-DnRun1) immediately preceding the first experience on the linear track and the recordings continued during the first run session on a novel track (DnRun1). Following their first run experience on the linear track, the animals were placed back in the sleep/rest box and allowed to sleep/rest (post-DnRun1 or pre-DnRun2), after which they were exposed for a second session of run on the same linear track (DnRun2). This run was followed by another session in the sleep/rest box (post-DnRun2). The first two (KO1 and KO2) out of the eight recorded mice were exposed for the third time to the linear track, followed by an additional session in the sleep/rest box. The remaining six mice went through, under the de novo condition, two run sessions (DnRun1 and DnRun2) each preceded and followed by a sleep/rest session. In the two KO mice that were exposed for a third session on the first novel track, the fields were not significantly more tuned and the lap-by-lap correlations were not as high in this session compared with their next day FamRun1 session or with the DnRun2 session in controls. There was no improved spatial tuning during the additional run session in the first two KO mice. In one KO animal (mouse KO4), no spiking events could be detected during de novo sleep/rest sessions due to the below threshold number of synchronously active cells.

In the contiguous condition, following a recording session on the now familiar linear track (FamRun1), a barrier that was blocking access to a contiguous novel linear track was lifted and the animals explored the L-shaped linear track for the first time (ContigRun). The orientation of the L-shaped track in the room and the room landmarks were kept constant throughout the experiment. Sessions in the sleep/rest box preceded (pre-ContigRun; before the barrier was lifted) and followed ContigRun (post-ContigRun; after the barrier was lifted). In one CT animal (mouse CT1), no spiking events were detected during pre-ContigRun due to the below threshold number of synchronously active cells. In three KO (mice KO1, KO2, and KO4) and two CT animals (mice CT1 and CT4), FamRun1 was recorded after an overnight sleep, whereas FamRun1 was recorded several hours after the de novo exposure to the novel track in the remaining animals. In three animals (CT4, KO1, and KO4) the pre-ContigRun sleep/rest session was preceded by a run session on the now familiar track, fRun (*Table 1*).

In the Disjunct condition, the animals (three CT and three KO mice, *Table 1*) were re-exposed to the now familiar L-shaped track 2 days later, after which they were allowed to sleep/rest in the sleep/rest

box (pre-DisjRun1 session). Subsequently, they explored an additional linear track on the same maze apparatus in isolation, separated by barriers at both ends from any familiar track, for two run sessions (DisjRun1 and DisjRun2) separated by a sleep/rest session in the sleep/rest box (post-DisjRun1 or pre-DisjRun2). All of the Run data were collected while the animals ran on the tracks (velocity of animal's movement higher than 5 cm/s), whereas all sleep/rest data were collected while animals were in the sleep/rest box (velocity less than 1 cm/s, and overwhelmingly 0 cm/s).

## Recordings and single unit analysis

A total of 458 neurons were recorded from the CA1 area of the hippocampus in four CT and four KO mice across the experimental sessions. Of these, 69 neurons in CT (13, 20, 26, and 10 in CT1–4) and 74 neurons in KO mice (20, 27, and 27 in KO1–3) were recorded in the de novo condition, 75 neurons in CT (25, 23, and 27 in CT2–4) and 88 neurons in KO (18, 25, 24, and 21 in KO1–4) were recorded in the Contig condition, whereas 76 neurons in CT (26, 23, and 28 in CT2–4) and 77 neurons in KO (20, 29, and 28 in KO1, 3–4) were recorded in the disjunct condition. Single cells were identified and place fields were computed as described earlier (*Dragoi and Tonegawa, 2011*). Spatial information was calculated for each individual cell in non-overlapping 2 cm spatial bins as described earlier (*Skaggs et al., 1996*) and the average values within-sessions/conditions were normalized for each genotype by the average values during FamRun1 (de novo and contiguous conditions) and FamRun2 (disjunct condition) to compare the across session changes in both genotypes.

## Preplay and replay analyses

To analyze preplay and replay processes, spiking events were detected during pre- and post-Run sleep/rest periods in the sleep/rest box in all experimental conditions. A spiking event (*Dragoi and Tonegawa, 2011*) was defined as a transient increase in the firing activity of a population of at least five different place cells within a temporal window preceded and followed by at least 100 ms of silence. For all conditions, the spikes from all the place cells active during run that were emitted during the preceding and following sleep/rest were sorted by time and further used for the detection of the spiking events. For the calculation of the temporal sequence, the times of the first spike emitted by each of the cells participating in the spiking event were sorted to determine the temporal order of neuronal firing (*Diba and Buzsaki, 2007*; *Dragoi and Tonegawa, 2013*). All four CT and three KO animals exhibited a significant number of spiking events in the sleep/rest sessions of the de novo condition, three CT and four KO animals exhibited a significant number of spiking events in the contig condition, whereas all three CT and three KO mice exhibited a significant number of spiking events in the Disjunct condition. The remaining animals had a below threshold number of simultaneously active CA1 place cells. The place cell sequences (templates) were calculated for each direction of the animal's movement and for each run session in all experimental conditions by ordering the spatial location of the place field peaks that were above 1 Hz. For place cells with multiple place fields above 1 Hz on a particular arm or track in the contiguous condition, only the place field corresponding to the peak firing rate of the place cell on that arm or track was considered for the construction of the template of that particular arm or track. This method is consistent with previous studies that employed spatial templates to demonstrate replay (*Lee and Wilson, 2002*; *Foster and Wilson, 2006*; *Diba and Buzsaki, 2007*) and preplay (*Dragoi and Tonegawa, 2011*) during sleep or awake rest. Place cells with fields on both the novel arm in the ContigRun session and the familiar track in the FamRun1 session participated in the construction of both the novel arm and familiar track templates. Statistical significance was calculated for each event by comparing the rank-order correlation between the sequence of cells' firing in the event (i.e., event sequence) and the place cell sequence (template) and the distribution of correlation values between the event sequence and 100 surrogate templates obtained by shuffling the order of place cells (*Diba and Buzsaki, 2007*; *Dragoi and Tonegawa, 2011*). The significance level was set at 0.025 to control for multiple comparisons (i.e., the two directions of run). The proportions of significant events (preplay and replay) were calculated as the ratio between the number of significant events and the total number of spiking events in which at least five corresponding place cells were active (*Diba and Buzsaki, 2007*; *Dragoi and Tonegawa, 2011*). Ripple oscillations were detected during sleep/rest periods in the sleep/rest box. The EEG signal was filtered (120–200 Hz) and ripple-band amplitude was computed using the Hilbert transform. Ripple epochs with maximal amplitude higher than four standard deviations above the mean, beginning and ending at one standard deviation were detected.

The overall significance of the preplay or replay process was calculated by comparing the group of correlation values of all events relative to the original template with each of the 100 groups of an equal number of correlation values relative to the shuffled surrogate templates using the ranksum test. The highest p value out of the resulting 100 p values (the weakest significant level) is further reported (*Figure 4*), except for the contiguous condition in CT where the criterion of p<0.025 was applied due to smaller, yet significant, differences between the original data and the shuffles (*Figure 4E*).

## Stability of place cell maps

The stability of place cell firing on the novel track (de novo and disjunct conditions) and novel arm (contig condition) in the beginning vs the end of the run session were assessed by calculating, for each place cell and each direction, a correlation between the spatial firing in the corresponding paired situations (i.e., the first four laps vs the last four laps of the run on the novel track or arm [*Dragoi and Tonegawa, 2011*]). The place cell activity was not partitioned in place fields, rather the whole activity on the particular track or arm was considered separately for each cell and direction (average correlations are shown in *Figure 1E–G*). In addition, we performed the same type of correlations while shuffling the identity of the cell in one member of the correlation (once for each different cell). Shuffle results were computed as correlation between spatial tuning of cells on the novel arm (or novel track) during the beginning of ContigRun (or de novo and disjunct runs) and spatial tuning of all the other simultaneously recorded cells on the novel arm (or novel track) during the end of ContigRun (Novel arm group) or de novo and disjunct runs. Original and shuffled correlations were compared using the ranksum test. In all cases, the original correlations were significantly higher than the shuffled ones (shuffled correlations values were ~0.2 across conditions).

## Lap-by-lap co-variations in firing rate

For the calculation of lap-by-lap correlations, spike times of each place field (velocity >5 cm/s) were binned at 3 s. After excluding the common zero-value bins, a correlation coefficient was calculated between the binned activities of pair members for all place field pairs (*Dragoi and Buzsaki, 2006*). This measure reflects the degree of co-variation of neuron pairs on multiple laps (trials), and its value is strongly correlated with the theta timescale temporal correlation of pairs of neurons (*Dragoi and Buzsaki, 2006*). The absolute value of significant correlation values (p<0.05) were compared across sessions and genotype. The mice ran 12–18 laps/session, similar numbers across genotypes (p>0.5, ranksum test). The duration of each session (trial) is entered in the *Table 1*. For the ContigRun session of the contiguous condition only, we could separate the place fields active on the familiar arm from those active on the novel arm. We calculated three sets of correlation values, between pairs of place fields that were: (1) Both expressed on the familiar arm (Fam × Fam), (2) Both expressed on the novel arm (Novel × Novel), and (3) Expressed one on the familiar arm and one on the novel arm (Fam × Novel).

## Comparison between features of temporal sequences

Features of temporal sequences were compared across genotypes using a paired *t* test applied to grand averages of parameters computed on data recorded during eight sleep/rest sessions: pre-DnRun1, post-DnRun1, pre-DnRun2, post-DnRun2, pre-ContigRun, post-ContigRun, pre-DisjRun1, and post-DisjRun1. For each genotype, the data were calculated for each sleep/rest session for each individual mouse as well as by averaging the specific parameters collected from all of the corresponding mice.

## T-maze alternation behavior

The behavioral data were collected from a total of 12 mice performing a delayed alternation task in two configurations of a T-maze. The size of the T-maze apparatus was 90 × 90 cm and the 3-D dimensions of the linear tracks were configured as in the neurophysiology part of the experiment. The mice were food deprived over 1 week to 85% of their body weight and were further trained from naïve state for ten days to alternate between the two arms of T-maze 1 (*Figure 6A* inset, vertical arms) for food rewards placed at each arm end. No barriers were ever used in the choice phase throughout training. On the return from the end of the choice arm to the start point located at the free end of the stem arm (*Figure 6A* inset, horizontal arm), a temporary barrier blocked animal access to the other choice arm. The criterion for learning was set at 70% correct choices per session for 2 consecutive days for each group of mice (i.e., KO and CT). Each mouse was trained for one session of ~20 trials each day. At the

end of the 10 days over which both groups reached the criterion, all mice were exposed to a second configuration of the T-maze (T-maze 2) for 2 additional days. From the beginning to the end of each trial over the 12 days, mice behave freely in T-mazes: they self-initiated their first trial, made a choice of an arm, returned to the start point, and self-initiated the next trial, for ~20 trials/session/day. For each day of the experiment and for each genotype, the performance of all animals was averaged and entered as a data point (*Figure 6*). For the first 10 days of the experiment, data were grouped in five blocks of two consecutive days and analyzed using a balanced two-way ANOVA test followed by the ranksum test.

## Acknowledgements

We thank MA Wilson for assistance with data acquisition and comments on an earlier version of the manuscript.

## Additional information

### Funding

| Funder | Grant reference number | Author |
| --- | --- | --- |
| National Institutes of Health | R01-MH078821, P50-MH58880 | Susumu Tonegawa |
| RIKEN BSI | | Susumu Tonegawa |

The funders had no role in study design, data collection and interpretation, or the decision to submit the work for publication.

### Author contributions

GD, Conception and design, Acquisition of data, Analysis and interpretation of data, Drafting or revising the article; ST, Conception and design, Drafting or revising the article

### Ethics

Animal experimentation: This study was performed in strict accordance with the recommendations in the Guide for the Care and Use of Laboratory Animals of the National Institutes of Health. All of the animals were handled according to approved institutional animal care and use committee of the Massachusetts Institute of Technology. The protocol was approved by the DCM Division of the Massachusetts Institute of Technology (protocol number 0112-004-15). All surgery was performed under Avertin anesthesia, and every effort was made to minimize suffering.

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
