## [Decision Letter]

Thank you for sending your work entitled “Development of hippocampal place cell assemblies during novel spatial experiences” for consideration at *eLife*. Your article has been favorably evaluated by a Senior editor and 3 reviewers, one of whom is a member of our Board of Reviewing Editors.

The Reviewing editor and the other reviewers discussed their comments before we reached this decision, and the Reviewing editor has assembled the following comments to help you prepare a revised submission. You will find below a large number of comments/critiques that are provided because the reviewers felt that aspects of the work were incompletely described or difficult to understand. We hope that these comments can help you improve the clarity, utility, and the logic of the final manuscript.

1) What is the relationship between place field instability and preplay? Lack of evidence for preplay could occur for two reasons, either: offline sequences of the target spatial representation were not generated, or the template shifted during learning (e.g., remapping) so that attempts to identify sequence matches are frustrated by changes in the template with experience. We know that in control mice, spatial firing changes more during a first experience than in KOs, therefore, it may be important to construct multiple templates for the first session, one early and one late, with the assumption that preplay would be equivalent in controls and KOs when the early template is used though preplay would be diminished in the late template due to CA3 NMDAR dependent plasticity that altered the order of place fields, as suggested in the cartoon in Figure 7.

2) The authors assert that their “data have shown that a novel representation of a first-time experience on linear tracks (DnRun1) is formed in the CA1 area primarily on the framework of the preconfigured hippocampal network”. As argued, if place fields were unimodal, then indeed a great deal of the structures of the spatial representation would be predetermined; however, based on the examples provided, there seems to be a great deal of firing that is out of order, so-to-speak. According to the model set forth by the authors, this out-of-order firing ought to depend upon plasticity incurred during experience, and therefore may be less stable in control mice as compared to CA3 KOs.

3) It is not clear what is meant by homeostasis with regard to preplay, replay, and learning. Either the authors should provide a more clear and concrete definition of this new use of the preplay, or leave it out altogether and fully embrace a variant of the authors' hypothesis that “temporal sequences emitted during onsite resting periods do not specifically reflect the recent spatial experience of the animal, but rather reflect a uniform distribution of multiple spatial experiences the animal had”. Why must the structure be dictated by a uniform generic distribution rather than reflecting recall of specific, related experiences (see McKenzie and Eichenbaum, Neuron 2011 and other recent commentaries on memory updating, e.g., Dudai)? Along the same line, the authors also argue that the rapid tuning of firing fields in control mice is evidence of schema modification. The authors should also consider a highly relevant experimental report (McKenzie et al., J Neurosci 2013) that specifically addressed how a neural correlate of memory changes with new related learning and the experimental conditions necessary for addressing the neural basis of schema modification.

4) Unlike the rest of the manuscript, one of the reviewers found the Abstract to be imprecisely written and difficult to understand, so much so that he/she was inspired to edit it as follows:

“Novel hippocampal spatial representations were thought to form during novel experiences and be specifically replayed the following sleep/rest enabling memory consolidation. We found that in naïve and experienced mice, novel CA1 place cell sequences exhibit similar correlations with their firing sequences during sleep/rests following and preceding the run, indicating network stability homeostasis. The tuning and spatial reliability coordination of place cell ensembles on novel tracks were reduced in naïve animals and showed accelerated increases in compared to experienced mice ones. Genetic blockade of CA3 NMDAR-dependent plasticity resulted in more rigid temporal-spatial sequences between sleep and exploratory behaviors and delayed tuning and coordination of place cell ensembles. These results indicate that novel spatial representations develop on the framework of preconfigured hippocampal networks whose homeostatic adjustment reorganization is accelerated by prior experience and CA3 NMDAR-dependent activity plasticity. The increased place map stability, tuning, and cell-assembly expression coordination may underlie rapid memory consolidation of repeated/related new experiences, but not of entirely novel ones.”

The use of several terms in the Abstract and in the main text seems imprecise, inappropriate or requires better explanation. Specifically, what is meant by homeostasis? Why is this not merely stability or inertia? What evidence is there that homeostasis is occurring? Do the authors argue that the mutants show greater homeostasis? The phrase “homeostatic reorganization” seems to be an oxymoron and no evidence of reorganization is presented; instead plenty evidence of mild adjustment and modifications of the firing sequences are demonstrated. The authors demonstrate the dependence of the temporal sequence and place cell tuning changes on CA3 NMDARs and presumably their activity but not synaptic plasticity itself as stated in the Abstract and in the manuscript.

What is meant by ensemble “coordination” and how did the authors measure it?

5) One of the reviewers found the first part of the Results (“Prior experience- and CA3 NMDAR-dependent homeostatic changes in sequence correlations”) difficult to read and follow. This is because the logic of the different experiments and comparisons is not explained, and so the reader has to figure it out. In fact, this logic is revealed in the second part of the Results (“Prior experience and CA3 NMDARs accelerate CA1 place cell tuning and coordination in novel environments”). It is almost as if the first part was written after the second part and then moved to the start of the Results. This needs to be rewritten. The authors might even consider switching the order of the two Results sections, which may be more appropriate anyway.

6) Why were paired t tests used to compare the KO and CT data? How is this possible?

7) How many laps did the mice run and how long did it take them to complete a trial? It appears to have been completed in < 5 minutes. If this is correct then the authors need to take this into account and elaborate their discussion of how they interpret the NMDAR KO effects on within-trial effects because the expression of the disrupted NMDAR-dependent plasticity may have a longer time course.

8) The results indicate and the model (Figure 7) predicts that both the temporal and spatial discharge sequences of place cells are relatively stable. How does this fundamental view point account for the basic observations and reports that these firing patterns remap, i.e., change radically and unpredictably between environments whether novel or familiar? How do the findings and preplay/replay during sleep/rest account for the viewpoint that place cell firing emerges as a consequence of convergent inputs from spatially-tuned cells in the entorhinal cortex such as grid, border and head-direction cells? What predictions does the authors’ model make for preplay/replay in these cells? The discussion of these issues is inadequate but will be important to readers.

9) Materials and methods

9.1) The description of the recording environment misses essential details.

- Were the walls of the maze transparent or opaque? If opaque, did they have a uniform color?

- Which kind of external landmarks were visible to the mice while in the walled maze?

- Was the constant (vertical) arm always in the same position with regard to the room?

9.2) Spiking events

- Were place cell sequences (templates) calculated for all simultaneously recorded CA1 cells or only for that subset that showed a linear progression?

9.3) Analysis of spatial tuning

- Spatial tuning of CA1 place cells was measured by place field length. This is one aspect of spatial information content, but another, and perhaps more important aspect, is the proportion of spiking in a contiguous area (field) vs random spiking throughout the environment (in other words, spatial signal-to-noise). An additional measure for this, such as Skagg's spatial information content should be added to the results.

- In addition, how was the field length calculated in case of two or more place fields? What was the defined minimum size for a place field?

10) Results

10.1) Figure 6 is a key figure to understand what is actually going on in the experiment. It is not explicitly told in the paper how familiar and novel arms were aligned for calculation the lap-by-lap correlations between the arms, but I assume the reward ends were superimposed. This should be clarified in the Materials and methods.

A corresponding correlation should be calculated between the familiar (vertical) arm of the L-maze and the isolated single arm of the disjunct condition.

The fact that the Fam x Nov correlations were equally high/low as Fam x Fam correlations suggests that there was no real place field remapping in the novel arm. Rather it appears that the distance to the reward was the major determinant for place cell firing as seen in numerous previous studies with a linear track. The similarity between the two arms was further enhanced by the 10cm high walls (if not transparent) preventing the mouse to well see extramaze landmarks.

This finding raises a concern whether any other condition as DeNovo can be called exposure to novelty.

10.2) The message of the paper would be much more concrete if examples of typical place fields under different conditions were shown.

11) Discussion

The section beginning: “In the absence of this type of plasticity, the CA1 cellular assemblies are less affected by the animals' novel spatial experiences and maintain an increased correlation across different brain states, behaviors, and experiences”.

This is true for the DeNovo condition but is actually opposite to what was observed in the Fam2 vs Disj1 transition!

It will be important to make a distinction between the DeNovo condition what was true exposure to novelty from the contiguous and disjunct conditions, where the novelty was only partial, and judged from the correlations, conditions when no remapping took place.

The results would thus suggest two roles for CA3 NMDA receptors. First, during exposure to novelty they enable flexible reorganization of the CA1 network (“remapping”). Second, when animals are exposed to only partial novelty, they enable pattern completion and reinstatement of the original representation. The discussion on the differences in preplay vs replay between CA3 NMDA-KO vs control animals should be adjusted accordingly.

---

## [Author Response]

*1) What is the relationship between place field instability and preplay? Lack of evidence for preplay could occur for two reasons, either: offline sequences of the target spatial representation were not generated, or the template shifted during learning (e.g., remapping) so that attempts to identify sequence matches are frustrated by changes in the template with experience. We know that in control mice, spatial firing changes more during a first experience than in KOs, therefore, it may be important to construct multiple templates for the first session, one early and one late, with the assumption that preplay would be equivalent in controls and KOs when the early template is used though preplay would be diminished in the late template due to CA3 NMDAR dependent plasticity that altered the order of place fields, as suggested in the cartoon in*
Figure 7.

The place fields expressed during the early laps and late laps of runs on familiar tracks are correlated to a 0.6-0.7 value (revised Figure 1). The place fields expressed during de novo runs are more dynamic, and show a correlation of 0.4-0.5 (revised Figure 1); this correlation is lower than on familiar tracks, yet significantly higher than chance level correlation, which is around 0.2 (cell identity shuffles; Dragoi and Tonegawa, Nature 2011; Materials and methods). The reviewers are asking how this relative instability of place fields (i.e., correlations of 0.4-0.5) in the very first run session of naïve animals affects the correlation between the spatial and temporal sequences of place cells (i.e., preplay and replay), here referred to as “spatial-temporal correlations”. Moreover, inspired by our cartoon model of temporal-spatial sequences (revised Figure 7), they are asking whether the early and late spatial templates display different spatial-temporal correlation values with the temporal sequences during the preceding sleep.

In order to answer the latter question, we constructed spatial templates from the activity of place cells in the first four laps of run (early templates) and the last four laps (late templates), and correlated them with the temporal sequences recorded during the previous sleep (early and late preplay). We found that the population of early template spatial-temporal correlations was not different from the population of late template ones in both control and mutant mice (revised Figure 5—figure supplement 1). However, both early and late spatial-temporal correlations were higher in the KO mice compared with the control ones (revised Figure 5—figure supplement 1), consistent with our results in revised Figure 5. These results indicate that the relative instability of the place fields during the first run is associated with a process of mild “editing” of the early template as illustrated in revised Figure 7, rather than with a process of dramatic “remapping” into a new chart as hypothesized by the reviewers. The relatively high correlation between early and late place fields in the de novo condition (0.4–0.5) is consistent with this scenario. Please also note the majority of significant correlations (preplay and replay) do not reach an absolute value of 1; this means, for example, that an early template and an edited late template could both have significant spatial-temporal correlations of 0.7 value with the same spiking event from the preceding sleep. A remapped late template would have spatial-temporal correlations values comparable with the shuffled ones, ∼0.2. Finally, we have explained in more detail in a previous publication (Dragoi and Tonegawa, Nature 2011) that an insufficient density of place cells per unit of linear track along with an insufficient sampling of spiking events in the pre-run sleep session will prevent the detection of preplay events.

In conclusion, the relative instability of place fields in the de novo condition does not prevent the detection of preplay; moreover, this instability would equally affect the detection of preplay and replay (if analyses of preplay would be frustrated by place field instabilities, the same would be true for replay).

*2) The authors assert that their “data have shown that a novel representation of a first-time experience on linear tracks (DnRun1) is formed in the CA1 area primarily on the framework of the preconfigured hippocampal network”. As argued, if place fields were unimodal, then indeed a great deal of the structures of the spatial representation would be predetermined; however, based on the examples provided, there seems to be a great deal of firing that is out of order, so-to-speak. According to the model set forth by the authors, this out-of-order firing ought to depend upon plasticity incurred during experience, and therefore may be less stable in control mice as compared to CA3 KOs*.

A number of place cells have more than one place field per linear track (please note that not every local maxima in the spatial tuning curve is a significant peak). We calculated the proportion of cells with more than one place field per track to be about 12% on the novel arm in the contiguous condition in control mice (Dragoi and Tonegawa, Nature 2011). The presence of an additional field is a well-known feature of place cells and all previous studies on replay using template matching or cell-pair analysis dealt with it by choosing the peak or the centroid of the main place field. We believe in our data (revised Figure 3) the place cell sequences on the novel tracks are quite well structured and easy to be visualized by general journal readers. The stability of the “out of order” firing is captured in part by the correlations between place cell activity in the early and late laps (revised Figure 1) and is similar across genotypes in the de novo condition. In the first paragraph of the second section of the Discussion we indeed stated that novel representations are formed on the framework of the preconfigured network, but we also added that they are “modified in part during the experience”, as also suggested in our response to the first comment, above.

*3) It is not clear what is meant by homeostasis with regard to preplay, replay, and learning. Either the authors should provide a more clear and concrete definition of this new use of the preplay, or leave it out all together and fully embrace a variant of the authors' hypothesis that “temporal sequences emitted during onsite resting periods do not specifically reflect the recent spatial experience of the animal, but rather reflect a uniform distribution of multiple spatial experiences the animal had”. Why must the structure be dictated by a uniform generic distribution rather than reflecting recall of specific, related experiences (see McKenzie and Eichenbaum, Neuron 2011 and other recent commentaries on memory updating, e.g., Dudai)? Along the same line, the authors also argue that the rapid tuning of firing fields in control mice is evidence of schema modification. The authors should also consider a highly relevant experimental report (McKenzie et al., J Neurosci 2013) that specifically addressed how a neural correlate of memory changes with new related learning and the experimental conditions necessary for addressing the neural basis of schema modification*.

The reviewers are correct, preplay may not reflect a uniform distribution of the multiple experiences the animal had, but rather a more specific distribution of related ones. In fact, we used the argument of schema-based representations to explain the preplay activity seen in the contiguous and disjunct conditions because of animals’ recent similar experiences in the de novo condition. However, given that experimentally naïve animals had never run on long linear tracks (de novo condition animals), it is not clear what similar experiences does preplay reflect in the de novo condition. It is for this reason that we more generally referred to “multiple spatial experiences the animal had” rather than to “specific, related experiences”. We included the references to the relevant papers from the Eichenbaum lab; thank you for pointing those out. We have explained our choice of using the term homeostasis in two answers below (item 4).

*4) Unlike the rest of the manuscript, one of the reviewers found the Abstract to be imprecisely written and difficult to understand, so much so that he/she was inspired to edit it as follows*:

“Novel hippocampal spatial representations were thought to form during novel experiences and be specifically replayed the following sleep/rest enabling memory consolidation. We found that in naïve and experienced mice, novel CA1 place cell sequences exhibit similar correlations with their firing sequences during sleep/rests following and preceding the run, indicating network stability homeostasis. The tuning and spatial reliability coordination of place cell ensembles on novel tracks were reduced in naïve animals and showed accelerated increases in compared to experienced mice ones. Genetic blockade of CA3 NMDAR-dependent plasticity resulted in more rigid temporal-spatial sequences between sleep and exploratory behaviors and delayed tuning and coordination of place cell ensembles. These results indicate that novel spatial representations develop on the framework of preconfigured hippocampal networks whose homeostatic adjustment reorganization is accelerated by prior experience and CA3 NMDAR-dependent activity plasticity. The increased place map stability, tuning, and cell-assembly expression coordination may underlie rapid memory consolidation of repeated/related new experiences, but not of entirely novel ones.”

We thank the reviewer for editing the Abstract. We incorporated these changes/comments in our revised version of the Abstract (150-word limit).

*The use of several terms in the Abstract and in the main text seems imprecise, inappropriate or requires better explanation. Specifically, what is meant by homeostasis? Why is this not merely stability or inertia? What evidence is there that homeostasis is occurring? Do the authors argue that the mutants show greater homeostasis? The phrase “homeostatic reorganization” seems to be an oxymoron and no evidence of reorganization is presented; instead plenty evidence of mild adjustment and modifications of the firing sequences are demonstrated. The authors demonstrate the dependence of the temporal sequence and place cell tuning changes on CA3 NMDARs and presumably their activity but not synaptic plasticity itself as stated in the Abstract and in the manuscript*.

More generally, the term homeostasis defines “the ability of the body or a cell to seek and maintain a condition of equilibrium or stability within its internal environment when dealing with external changes”. In our study, the term homeostasis refers to the ability of the hippocampal network (and associated brain areas) to maintain the stability of temporal sequences expressed during sleep before and after “dealing” with a novel run experience: we report similar incidence of significant p/replay events and similar spatial-temporal correlation strengths pre- and post-run experience. The term inertia proposed by the reviewers seem to refer more to a physical rather than a biological system; it might somehow capture the representational processes in the CA3 NMDAR KO mice, although for that case we prefer to use the terms rigidity and less flexible. We have uncoupled the terms homeostatic and reorganization, and substituted the term reorganization with the terms modification and adjustment. We have substituted the expression NMDAR-dependent plasticity with NMDAR-dependent activity. We thank the reviewers for pointing out these points.

*What is meant by ensemble “coordination” and how did the authors measure it*?

The term ensemble coordination refers to the tendency of groups of cells that are members of a cell assembly to co-vary their firing rates across individual running laps (also called lap-by-lap co-variation or correlation throughout the manuscript). We measure this coordination by calculating the correlation coefficient between the firing rates binned across an entire run session, for pairs of cells. The procedure has been described in detail in Dragoi and Buzsaki, Neuron, 2006 and more succinctly in the current Materials and methods section. We have replaced the term coordination with co-variation throughout the manuscript, a term that we first introduced in the Results section.

*5) One of the reviewers found the first part of the Results (“Prior experience- and CA3 NMDAR-dependent homeostatic changes in sequence correlations”) difficult to read and follow. This is because the logic of the different experiments and comparisons is not explained, and so the reader has to figure it out. In fact, this logic is revealed in the second part of the Results (“Prior experience and CA3 NMDARs accelerate CA1 place cell tuning and coordination in novel environments”). It is almost as if the first part was written after the second part and then moved to the start of the Results. This needs to be rewritten. The authors might even consider switching the order of the two Results sections, which may be more appropriate anyway*.

We have reversed the order in which present our Results as suggested by the reviewers.

*6) Why were paired t tests used to compare the KO and CT data? How is this possible*?

The comparisons between genotypes reported were made between groups of average values from each session for each genotype, paired by session type (displayed in the revised Figure 5); we used this analysis because the independent variable being tested in those cases was the genotype. The average values reported are grand averages; we clarified this in the Materials and methods section of the revised manuscript.

*7) How many laps did the mice run and how long did it take them to complete a trial? It appears to have been completed in < 5 minutes. If this is correct then the authors need to take this into account and elaborate their discussion of how they interpret the NMDAR KO effects on within-trial effects because the expression of the disrupted NMDAR-dependent plasticity may have a longer time course*.

The mice ran 12–18 laps per session, similar numbers across genotypes (p>0.5, ranksum test). The duration of each session (trial) is entered in Table 1. The data from the run sessions are specifically in the columns containing the word ‘Run’. The sessions the reviewers appears to refer to (i.e., novel runs) are each longer than 15 minutes, and they average 40–45 minutes in the first session of de novo condition. The two sessions of the de novo condition (over 1 h altogether) provided ample time for the non-NMDAR-dependent plasticity to occur. The results suggest that CA3 NMDAR-dependent activity is specifically involved in the within-session changes in single and ensemble place cell activity. We have added this comment to the revised manuscript.

*8) The results indicate and the model (*Figure 7*) predicts that both the temporal and spatial discharge sequences of place cells are relatively stable. How does this fundamental view point account for the basic observations and reports that these firing patterns remap, i.e., change radically and unpredictably between environments whether novel or familiar? How do the findings and preplay/replay during sleep/rest account for the viewpoint that place cell firing emerges as a consequence of convergent inputs from spatially-tuned cells in the entorhinal cortex such as grid, border and head-direction cells? What predictions does the authors’ model make for preplay/replay in these cells? The discussion of these issues is inadequate but will be important to readers*.

We have recently investigated the relationship between preplay and multiple highly distinct firing patterns (i.e., remapping) using 3 novel linear tracks (Dragoi and Tonegawa, PNAS 2013). Briefly, in animals naïve to running on relatively long linear tracks (1.5 m each), the CA1 hippocampal network is capable of preplaying multiple distinct spatial firing patterns that will be expressed in future novel environments; it thus appears that this type of remapping, while significant, is not entirely unpredictable. Regarding the relative contribution of entorhinal layer III and of CA3 area inputs to the activity of CA1 place cells, several studies have now shown that these inputs are not necessary for the expression of place cells in the CA1, despite both of them sending convergent inputs onto CA1 (works from the Moser and the Tonegawa labs). The highly periodic pattern of grid cell activity and the rather persistent activity of border cells might not make them the best candidates to study p/replay in these cells, at least not by simply employing the conventional spatial-temporal sequence analysis that is being widely used for CA1 place cell sequences.

9) Materials and methods

*9.1) The description of the recording environment misses essential details*.

*- Were the walls of the maze transparent or opaque? If opaque, did they have a uniform color*?

The walls of the maze were translucent whereas the barriers were opaque, with uniform black color. We have added this information in the revised version.

*- Which kind of external landmarks were visible to the mice while in the walled maze*?

The walls were 10 cm high and the mice had the microdrive and the tether attached; their weight was counterbalanced by external remote weights via pulley systems. The mice had free visual access toward the ceiling and higher room landmarks such as ceiling lights, high curtains, and the ceiling itself.

*- Was the constant (vertical) arm always in the same position with regard to the room*?

Yes. We have added this information in the revised version.

9.2) Spiking events

*- Were place cell sequences (templates) calculated for all simultaneously recorded CA1 cells or only for that subset that showed a linear progression*?

All simultaneously recorded place cells that fired > 1 Hz peak rate on a particular track were ordered based on the location of their peak firing on that track, for each of the 2 directions, in order to construct the spatial templates.

9.3) Analysis of spatial tuning

*- Spatial tuning of CA1 place cells was measured by place field length. This is one aspect of spatial information content, but another, and perhaps more important aspect, is the proportion of spiking in a contiguous area (field) vs random spiking throughout the environment (in other words, spatial signal-to-noise). An additional measure for this, such as Skagg's spatial information content should be added to the results*.

In Figure 6 in the original submission (Figure 2 in the revised version) we displayed the spatial information content calculated as in Skaggs et al, Hippocampus, 1996. We also had this information in the Results section and in the Materials and methods section.

*In addition, how was the field length calculated in case of two or more place fields? What was the defined minimum size for a place field*?

In cases where cells had more than one field passing the threshold, all fields were included in the analysis. The average number of fields/cell is given in the Results section for each session. The minimum size for including a place field was the highest value between 10 cm and the length of track over which the cell was active at >10% of peak firing, as described in Dragoi and Tonegawa, Nature, 2011.

10) Results

*10.1)*
Figure 6
*is a key figure to understand what is actually going on in the experiment. It is not explicitly told in the paper how familiar and novel arms were aligned for calculation the lap-by-lap correlations between the arms, but I assume the reward ends were superimposed. This should be clarified in the Materials and methods*.

*A corresponding correlation should be calculated between the familiar (vertical) arm of the L-maze and the isolated single arm of the disjunct condition*.

The lap-by-lap coordination (co-variation) can only be calculated for pairs of cells active at the same time on the same track on multiple laps. For the ContigRun session of the contiguous condition only, we could separate the place fields active on the familiar arm from those active on the novel arm. We calculated three sets of correlation values, between pairs of place fields that were: 1. Both expressed on the familiar arm (Fam x Fam); 2. Both expressed on the novel arm (Novel x Novel); and 3. Expressed one on the familiar arm and one on the novel arm (Fam x Novel). We cannot calculate the coordination between place fields on the L-maze and those from cells active in the disjunct condition, because they occur at very different time points. We have explained in more detail this method in the Materials and methods section.

*The fact that the Fam x Nov correlations were equally high/low as Fam x Fam correlations suggests that there was no real place field remapping in the novel arm. Rather it appears that the distance to the reward was the major determinant for place cell firing as seen in numerous previous studies with a linear track. The similarity between the two arms was further enhanced by the 10 cm high walls (if not transparent) preventing the mouse to well see extramaze landmarks*.

*This finding raises a concern whether any other condition as DeNovo can be called exposure to novelty*.

As we indicate in the item above (10.1) the within-session lap-by-lap coordination measure does not assess the across-sessions remapping level. Since the changes in relative distance between pairs of place fields during runs on familiar versus novel tracks (FamRun1 versus ContigRun and FamRun2 versus DisjRun1) take relatively high values (>90% are larger than 6 cm and >50% are larger than 20 cm, revised Figure 2) we can reject the hypothesis that there is no remapping in the contiguous and disjunct conditions. On the contrary, there is no significant correlation between the order of place cell firing on the familiar arm and the one on the novel arm/track, in the contiguous/disjunct conditions, further demonstrating place field remapping across the different linear environments. We have added this comment into the revised version.

*10.2) The message of the paper would be much more concrete if examples of typical place fields under different conditions were shown*.

We have included examples of typical place fields under different experimental conditions (revised Figure 1).

11) Discussion

*The section beginning: “In the absence of this type of plasticity, the CA1 cellular assemblies are less affected by the animals' novel spatial experiences and maintain an increased correlation across different brain states, behaviors, and experiences”*.

*This is true for the DeNovo condition but is actually opposite to what was observed in the Fam2 vs Disj1 transition*!

Our claim is simply aiming to interpret the higher spatial-temporal correlation values (p/replay) observed in KO versus CT mice in all experimental conditions (including disjunct condition). The expression “different brain states, behaviors, and experiences” refers to sleep versus exploratory behaviors, when the brain state (ripples/slow wave sleep versus theta oscillations during run), the behavior, and the overall animal experience are indeed different. We have clarified this in the revised version.

*It will be important to make a distinction between the DeNovo condition what was true exposure to novelty from the contiguous and disjunct conditions, where the novelty was only partial, and judged from the correlations, conditions when no remapping took place*.

We are making a distinction between the different types of novelty given by the 3 conditions, but we respectfully disagree with the reviewers’ interpretation that there is no remapping in the contiguous and disjunct conditions, as explained above.

*The results would thus suggest two roles for CA3 NMDA receptors. First, during exposure to novelty they enable flexible reorganization of the CA1 network (“remapping”). Second, when animals are exposed to only partial novelty, they enable pattern completion and reinstatement of the original representation. The discussion on the differences in preplay vs replay between CA3 NMDA-KO vs control animals should be adjusted accordingly*.

For the reasons expressed above (10.1 and 11) we respectfully disagree with the reviewers’ reinterpretation of our data.